# Transformation of a temporal speech cue to a spatial neural code in human auditory cortex

Neal P Fox[1], Matthew Leonard[1], Matthias J Sjerps[2,3], Edward F Chang[1,4]*

[1]Department of Neurological Surgery, University of California, San Francisco, San Francisco, United States; [2]Donders Institute for Brain, Cognition and Behaviour, Centre for Cognitive Neuroimaging, Radboud University, Nijmegen, Netherlands; [3]Max Planck Institute for Psycholinguistics, Nijmegen, Netherlands; [4]Weill Institute for Neurosciences, University of California, San Francisco, San Francisco, United States

**Abstract** In speech, listeners extract continuously-varying spectrotemporal cues from the acoustic signal to perceive discrete phonetic categories. Spectral cues are spatially encoded in the amplitude of responses in phonetically-tuned neural populations in auditory cortex. It remains unknown whether similar neurophysiological mechanisms encode temporal cues like voice-onset time (VOT), which distinguishes sounds like /b/ and /p/. We used direct brain recordings in humans to investigate the neural encoding of temporal speech cues with a VOT continuum from /ba/ to /pa/. We found that distinct neural populations respond preferentially to VOTs from one phonetic category, and are also sensitive to sub-phonetic VOT differences within a population's preferred category. In a simple neural network model, simulated populations tuned to detect either temporal gaps or coincidences between spectral cues captured encoding patterns observed in real neural data. These results demonstrate that a spatial/amplitude neural code underlies the cortical representation of both spectral and temporal speech cues.

*For correspondence: edward.chang@ucsf.edu

Competing interests: The authors declare that no competing interests exist.

## Introduction

During speech perception, listeners must extract acoustic cues from a continuous sensory signal and map them onto discrete phonetic categories, which are relevant for meaning (*Stevens, 2002*; *Liberman et al., 1967*). Many such cues to phonological identity are encoded within the fine temporal structure of speech (*Shannon et al., 1995*; *Rosen, 1992*; *Klatt, 1976*). For example, voice-onset time (VOT), defined as the interval between a stop consonant's release and the onset of vocal fold vibration (acoustically, the *burst* and the *voicing*), is a critical cue that listeners use to distinguish *voiced* (e.g., /b/, /d/, /g/) from *voiceless* (e.g., /p/, /t/, /k/) stop consonants in English (*Liberman et al., 1958*; *Lisker and Abramson, 1964*). When the burst and voicing are roughly coincident (short VOT; ~0 ms), listeners perceive a bilabial stop as a /b/, but when voicing follows the burst after a temporal gap (long VOT; ~50 ms), listeners hear a /p/.

Recent evidence from human electrocorticography (ECoG) has shown that information about a speech sound's identity is encoded in the amplitude of neural activity at phonetically-tuned cortical sites in the superior temporal gyrus (STG) (*Mesgarani et al., 2014*). Distinct neural populations in this region respond selectively to different classes of phonemes that share certain spectral cues, such as the burst associated with stop consonants or the characteristic formant structure of vowels produced with specific vocal tract configurations. However, it is unclear whether phonetic categories distinguished by temporal cues (e.g., voiced vs. voiceless stops) are represented within an analogous spatial encoding scheme. If so, this would entail that local neural populations are tuned to detect

not merely the presence of certain spectral cues (the burst and voicing), but also their timing relative to one another.

In addition to distinguishing phonetic categories, the exact VOT of a given utterance of a /b/ or a /p/ will vary considerably depending on numerous factors such as speech rate, phonetic context, and speaker accent (*Miller et al., 1986*; *Kessinger and Blumstein, 1997*; *Klatt, 1975*; *Lisker and Abramson, 1967*; *Allen et al., 2003*; *Flege and Eefting, 1986*; *Fox et al., 2015*). Although only categorical phonetic identity (e.g., whether a particular VOT is more consistent with a /b/ or a /p/) is strictly necessary for understanding meaning, sensitivity to fine-grained sub-phonetic detail (e.g., whether a particular /p/ was pronounced with a 40 ms vs. a 50 ms VOT) is also crucial for robust speech perception, allowing listeners to flexibly adapt and to integrate multiple cues to phonetic identity online in noisy, unstable environments (*Miller and Volaitis, 1989*; *Clayards et al., 2008*; *Kleinschmidt and Jaeger, 2015*; *McMurray and Jongman, 2011*; *Toscano and McMurray, 2010*; *Fox and Blumstein, 2016*). However, the neurophysiological mechanisms that support listeners' sensitivity (*Kuhl, 1991*; *Carney, 1977*; *Pisoni and Tash, 1974*; *Massaro and Cohen, 1983*; *Andruski et al., 1994*; *McMurray et al., 2002*; *Schouten et al., 2003*) to such detailed speech representations are not known. We tested whether sub-phonetic information might be encoded in the neural response amplitude of the same acoustically-tuned neural populations that encode phonetic information in human auditory cortex.

To address these questions, we recorded neural activity directly from the cortex of seven human participants using high-density ECoG arrays while they listened to and categorized syllables along a VOT continuum from /ba/ (0 ms VOT) to /pa/ (50 ms VOT). We found that the amplitude of cortical responses in STG simultaneously encodes both phonetic and sub-phonetic information about a syllable's initial VOT. In particular, spatially discrete neural populations respond preferentially to VOTs from one category (either /b/ or /p/). Furthermore, peak response amplitude is modulated by stimulus VOT within each population's preferred – but not its non-preferred – voicing category (e.g., stronger response to 0 ms than to 10 ms VOT in voiced-selective [/b/-selective] neural populations). This same encoding scheme emerged in a computational neural network model simulating neuronal populations as leaky integrators tuned to detect either temporal coincidences or gaps between distinct spectral cues. Our results provide direct evidence that phonetic and sub-phonetic information carried by VOT are represented within spatially discrete, phonetically-tuned neural populations that integrate temporally-distributed spectral cues in speech. This represents a crucial step towards a unified model of cortical speech encoding, demonstrating that both spectral and temporal cues and both phonetic and sub-phonetic information are represented by a common (spatial) neural code.

## Results

Participants listened to and categorized speech sounds from a digitally synthesized continuum of consonant-vowel syllables that differed linearly only in their voice-onset time (VOT) from /ba/ (0 ms VOT) to /pa/ (50 ms VOT). This six-step continuum was constructed by manipulating only the relative timing of the spectral burst and the onset of voicing while holding all other acoustic properties of the stimuli constant (*Figure 1A/B*; see Materials and methods) (*Klatt, 1980*). Analysis of participants' identification behavior confirmed that stimuli with longer VOTs were more often labeled as /pa/ (mixed effects logistic regression: $\beta_{\text{VOT}} = 0.19$, $t = 17.78$, p=5.6*10$^{-63}$; data for example participant in *Figure 1C*; data for all participants in *Figure 1—figure supplement 1*). Moreover, and consistent with past work, listeners' perception of the linear VOT continuum was sharply non-linear, a behavioral hallmark of categorical perception (*Liberman et al., 1957*; *Liberman et al., 1961*; *Kronrod et al., 2016*). A psychophysical category boundary between 20 ms and 30 ms divided the continuum into stimuli most often perceived as voiced (/b/: 0 ms, 10 ms, 20 ms VOTs) or as voiceless (/p/: 30 ms, 40 ms, 50 ms VOTs).

### Temporal cues to voicing category are encoded in spatially distinct neural populations

To investigate neural activity that differentiates the representation of speech sounds based on a temporal cue like VOT, we recorded high-density electrocorticography in seven participants while they listened to the VOT continuum. We examined high-gamma power (70–150 Hz) (*Chang, 2015*; *Crone et al., 2001*; *Steinschneider et al., 2008*; *Ray and Maunsell, 2011*), aligned to the acoustic

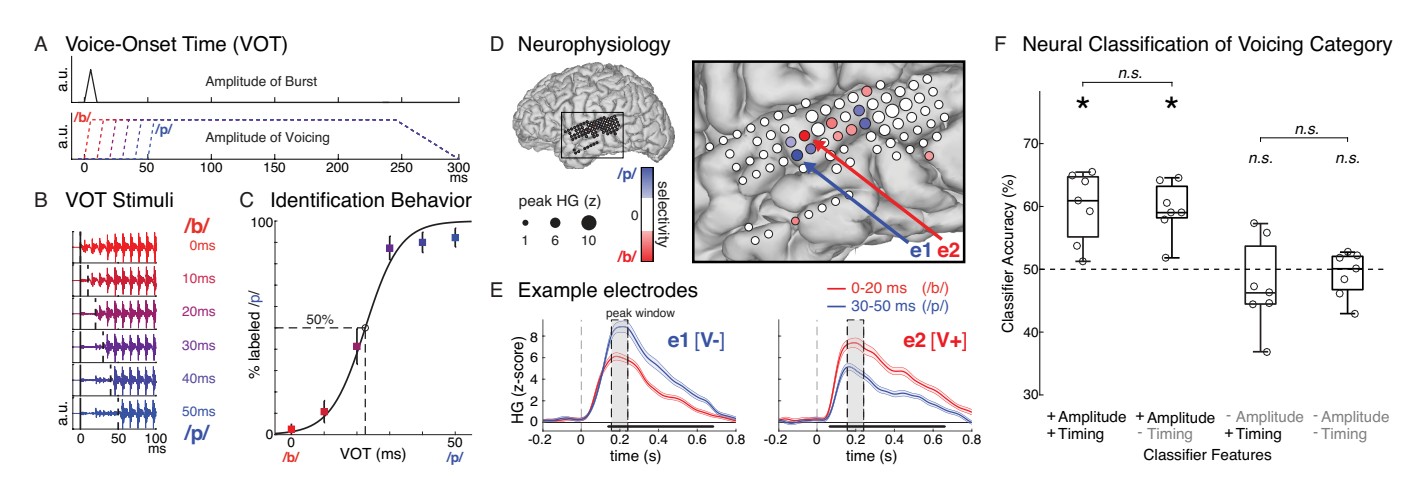

**Figure 1.** Speech sound categories that are distinguished by a temporal cue are spatially encoded in the peak amplitude of neural activity in distinct neural populations. (A) Stimuli varied only in voice-onset time (VOT), the duration between the onset of the burst (top) and the onset of voicing (bottom) (a.u. = arbitrary units). (B) Acoustic waveforms of the first 100 ms of the six synthesized stimuli. (C) Behavior for one example participant (mean ± bootstrap SE). Best-fit psychometric curve (mixed effects logistic regression) yields voicing category boundary between 20–30 ms (50% crossover point). (D) Neural responses in the same representative participant show selectivity for either voiceless or voiced VOTs at different electrodes. Electrode size indicates peak high-gamma (HG; z-scored) amplitude at all speech-responsive temporal lobe sites. Electrode color reflects strength and direction of selectivity (Spearman's ρ between peak HG amplitude and VOT) at VOT-sensitive sites (p<0.05). (E) Average HG responses (± SE) to voiced (0–20 ms VOTs; red) and voiceless (30–50 ms VOTs; blue) stimuli in two example electrodes from (D), aligned to stimulus onset (e1: voiceless-selective, V-; e2: voiced-selective, V+). Horizontal black bars indicate timepoints with category discriminability (p<0.005). Grey boxes mark average peak window (± SD) across all VOT-sensitive electrodes (n = 49). (F) Population-based classification of voicing category (/p/ vs. /b/) during peak window (150–250 ms after stimulus onset). Chance is 50%. Boxes show interquartile range across all participants; whiskers extend to best- and worst-performing participants; horizontal bars show median performance. Asterisks indicate significantly better-than-chance classification across participants (p<0.05; n.s. = not significant). Circles represent individual participants.

The online version of this article includes the following figure supplement(s) for figure 1:

**Figure supplement 1.** Identification behavior across all participants with behavioral data.

**Figure supplement 2.** Locations of all speech-responsive and VOT-sensitive electrodes in each participant (P1–P7).

**Figure supplement 3.** Analysis of evoked local field potentials reveals that some electrodes that encode VOT in their peak high-gamma amplitude also exhibit amplitude and/or temporal response features that are VOT-dependent.

**Figure supplement 4.** Complex and variable associations between VOT and amplitude/temporal features of auditory evoked local field potentials (AEPs) exist in responses of electrodes that robustly encode voicing in their peak high-gamma amplitude.

onset of each trial (burst onset), at every speech-responsive electrode on the lateral surface of the temporal lobe of each patient (n = 346 electrodes; see Materials and methods for details of data acquisition, preprocessing, and electrode selection).

We used nonparametric correlation analysis (Spearman's ρ) to identify electrodes where the peak high-gamma amplitude was sensitive to stimulus VOT. Across all participants, we found 49 VOT-sensitive sites, primarily located over the lateral mid-to-posterior STG, bilaterally. Peak response amplitude at these VOT-sensitive electrodes reliably discriminated between voicing categories, exhibiting stronger responses to either voiced (/b/; VOT = 0–20 ms; n = 33) or voiceless (/p/; VOT = 30–50 ms; n = 16) stimuli (*Figure 1D*; locations of all sites shown in *Figures 2A* and *Figure 1—figure supplement 2*). We observed that, within individual participants, electrodes spaced only 4 mm apart showed strong preferences for different voicing categories, and we did not observe any clear overall regional or hemispheric patterns in the prevalence or selectivity patterns of VOT-sensitive electrodes (see Materials and methods for additional information).

Robust category selectivity in voiceless-selective (V-) and voiced-selective (V+) neural populations emerged as early as 50–150 ms post-stimulus onset and often lasted for several hundred milliseconds (example electrodes in *Figure 1E*). Across all VOT-sensitive electrodes, voicing category selectivity was reliable whether a trial's voicing category was defined based on the psychophysically-determined category boundary (0–20 ms vs. 30–50 ms VOTs; V- electrodes: $z = 3.52$, $p=4.4\times10^{-4}$;

V+ electrodes: $z = -5.01$, p=5.4×10$^{-7}$; Wilcoxon signed-rank tests) or based on the actual behavioral response recorded for each trial (V- electrodes: p=4.9×10$^{-4}$; V+ electrodes: p=6.1×10$^{-5}$; Wilcoxon signed-rank tests).

These results show that spatially distinct neural populations in auditory cortex are tuned to speech sound categories defined by a temporal cue. Critically, if individual neural populations only responded to spectral features (e.g., to the burst or to the onset of voicing), we would not have observed overall amplitude differences in their responses to /b/ versus /p/ categories.

Given this pattern of spatial tuning, we tested whether the voicing category of single trials could be reliably decoded from population neural activity across electrodes. For each participant, we trained a multivariate pattern classifier (linear discriminant analysis with leave-one-out cross validation) to predict trial-by-trial voicing category using high-gamma activity across all speech-responsive electrodes on the temporal lobe during the peak neural response (150–250 ms after stimulus onset; see Materials and methods). We found that, across participants, classification accuracy was significantly better than chance (Wilcoxon signed-rank test: p=0.016; *Figure 1F*, leftmost box plot), demonstrating that spatially and temporally distributed population neural activity during the peak response contains information that allows for decoding of a temporally-cued phonetic distinction in speech.

## Peak neural response amplitude robustly encodes voicing category

Next, we asked which features of the population neural response encode voicing category. Specifically, we evaluated three alternatives for how temporally-cued voicing category is encoded by high-gamma responses in cortex during the peak neural response: (1) the spatial pattern of peak response amplitude across electrodes, (2) the temporal patterns of evoked responses across electrodes during the peak response, or (3) both amplitude and timing of neural activity patterns. We tested these hypotheses by selectively corrupting amplitude and/or temporal neural features that were inputs for the classifier. As with the previous analyses, and following prior work on speech sound encoding (*Mesgarani et al., 2014*), these analyses (*Figure 1F*) focused on cortical high-gamma activity during the peak response window (150–250 ms after stimulus onset; but see *Figure 3* for analyses of an earlier time window).

To corrupt temporal information, we randomly jittered the exact timing of the neural response for each trial by shifting the 100 ms analysis window by up to ±50 ms. Because the uniform random jitter was applied independently to each trial, this procedure disrupts any temporal patterns during the peak neural response that might reliably distinguish trials of different voicing categories, such as precise (millisecond-resolution) timing of the peak response at an electrode or the dynamics of the evoked response during the peak window, including *local* temporal dynamics (during a single electrode's peak response) or *ensemble* temporal dynamics (the relative timing of responses of spatially-distributed electrodes in the same participant). To corrupt amplitude information, we eliminated any condition-related differences in the peak response amplitude at every electrode. For each electrode, the evoked high-gamma response to all trials within a given voicing category were renormalized so that the average responses to both voicing categories had identical amplitudes at the peak, but could still vary reliably in the timing and dynamics during the peak window. These techniques allowed us to examine the relative contributions of temporal and amplitude information contained within the peak neural response window to the classification of voicing category (see Materials and methods for detailed description of this approach).

Across participants, we found that, when the classifiers had access to amplitude information but not timing information (+Amplitude/-Timing) during the peak response, performance was significantly better than chance (Wilcoxon signed-rank test: p=0.016; *Figure 1F*). Furthermore, despite the profound corruption of temporal information in the neural responses, classification accuracy was statistically comparable to the model that had access to both amplitude and timing information (+Amplitude/+Timing; Wilcoxon signed-rank test: p=0.69; *Figure 1F*), suggesting that amplitude information alone is sufficient for classifying a trial's voicing category.

In contrast, when amplitude information was corrupted and only temporal patterns in the peak response window were reliable (-Amplitude/+Timing), classifier performance was not different from chance (Wilcoxon signed-rank test: p=0.69; *Figure 1F*) and was worse for every participant compared to the model with both types of information (Wilcoxon signed-rank test: p=0.016). Finally, we compared the model with only timing information to a model where both amplitude and timing

information during the peak window were corrupted (-Amplitude/-Timing). We found that preserving timing information alone had no effect on classification performance compared to the most impoverished model (-Amplitude/-Timing; Wilcoxon signed-rank test: p=0.58; *Figure 1F*), which also failed to perform better than chance (Wilcoxon signed-rank test: p=0.94; *Figure 1F*). Together, these results constitute evidence for a spatial/amplitude code for speech categories that differ in a temporal cue. Thus, localized peak high-gamma response amplitude spatially encodes voicing of single trials in STG, analogous to other spectrally-cued phonetic features (*Mesgarani et al., 2014*). Note that, while spatial (and not temporal) patterns of high-gamma responses robustly encode voicing during this critical peak window, we later describe additional analyses that address possible temporal encoding patterns in the local field potential (*Figure 1—figure supplements 3* and *4*) and in an earlier time window (*Figure 3*).

The encoding of stop consonant voicing in the amplitude of evoked high-gamma responses in STG suggests that the representation of temporally-cued phonetic features may be explained within the same neural coding framework as the representation of spectrally-cued phonetic features. However, previous work on the cortical representation of voicing has identified a role for temporal information in the local field potential (LFP) (*Steinschneider et al., 1999*; *Steinschneider et al., 2013*), which is dominated by lower- frequencies (*Buzsáki et al., 2012*; *Einevoll et al., 2013*).

To link our results with this existing literature, we conducted a series of exploratory analyses of the neural responses to our stimuli using the raw voltage (LFP) signal. For each VOT-sensitive electrode (defined in the high-gamma analysis), we measured the correlations between VOT and peak latency and between VOT and peak amplitude for three peaks in the auditory evoked potential (AEP) occurring approximately 75–100 ms ($P_\alpha$), 100–150 ms ($N_\alpha$), and 150–250 ms ($P_\beta$) after stimulus onset (*Figure 1—figure supplement 3*; *Howard et al., 2000*; *Nourski et al., 2015*). We found that some VOT-sensitive electrodes encoded VOT in the latency of these peaks (e.g., *Figure 1—figure supplement 4*, panels E/I/M), replicating previous results (*Steinschneider et al., 2011*). However, among electrodes that encode VOT in peak high-gamma amplitude, there exist many more electrodes that *do not* encode VOT in these temporal features of the AEP, and many that also encode VOT in the amplitude of these AEP peaks (*Figure 1—figure supplements 3* and *4*). This further supports the prominent role that amplitude information plays in the neural representation of voicing and VOT, both in high-gamma and in the LFP. Therefore, subsequent analyses focus on the high-gamma amplitude. (For detailed descriptions of these LFP analyses and their results, see Methods and *Figure 1—figure supplements 3* and *4*).

## Peak response amplitude encodes sub-phonetic VOT information within preferred category

Next, we assessed whether VOT-sensitive neural populations (*Figure 2A*), which reliably discriminate between phonetic categories (voiced vs. voiceless), also encoded within-category sub-phonetic detail in the peak response amplitude. Specifically, the cortical representation of stimuli from the same voicing category but with different VOTs (e.g., 30, 40, and 50 ms VOTs that all correspond to /p/) could be either categorical (i.e., all elicit the same peak response amplitude) or graded (i.e., peak response amplitude depends on within-category VOT).

We examined the average responses to each of the six VOTs separately in the voiceless-selective electrodes (V-; *Figure 2B*) and the voiced-selective electrodes (V+; *Figure 2C*). We observed clear differences in activity evoked by different VOTs at the peak response (~200 ms after stimulus onset), even within the same voicing category, consistent with sensitivity to sub-phonetic detail (*Blumstein et al., 2005*; *Toscano et al., 2010*; *Toscano et al., 2018*; *Frye et al., 2007*). However, the discriminability of responses to within-category VOTs depended on the preferred voicing category of a given electrode.

To quantify this observation, at each electrode, we computed the rank-based correlation (Spearman's ρ) between stimulus VOT and peak response amplitude separately for each voicing category (0–20 ms and 30–50 ms VOTs). This procedure resulted in two correlation coefficients for each VOT-sensitive site ($\rho_{0-20}$, $\rho_{30-50}$) and corresponding test statistics reflecting the strength of within-category amplitude encoding of stimulus VOT in each voicing category. These test statistics (one per voicing category per VOT-sensitive electrode) then served as the input data for a series of signed-rank statistical tests to assess overall within-category encoding properties of groups of electrodes (e.g., of all V- electrodes) (see Methods for details). For example, consider V- electrodes, which exhibit stronger

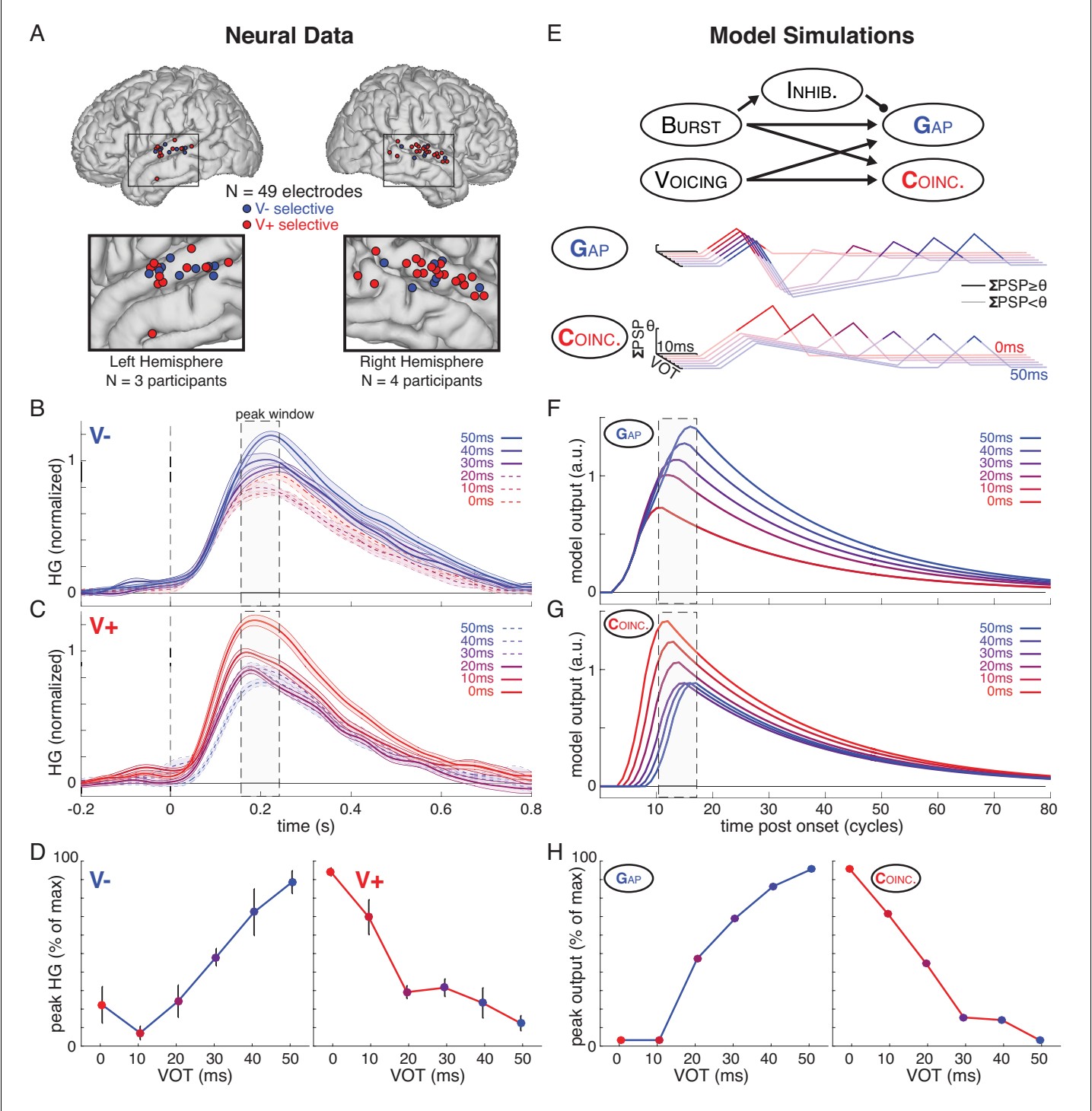

**Figure 2.** Human auditory cortex encodes both phonetic (between-category) and sub-phonetic (within-category) information in peak response amplitude, which can be modeled by a simple neural network that implements temporal gap and coincidence detection. (A) Spatial distribution of VOT-sensitive electrodes across all participants (on standardized brain). (B) Average (± SE) normalized HG response to each VOT across all voiceless-selective (V-) electrodes, aligned to stimulus onset. Line style denotes category membership of a given VOT (solid: preferred category; dashed: non-preferred category). Grey box marks average peak window (± SD) across all VOT-sensitive electrodes. (C) Average (± SE) normalized response to each VOT across all voiced-selective (V+) electrodes. (D) Average (± SE) peak response to each VOT stimulus for V- electrodes (left) and V+ electrodes (right) (see Materials and methods). (E) A simple neural network model (top) comprised of five leaky integrator nodes was implemented to examine computational mechanisms that could account for the spatial encoding of a temporal cue (VOT). Arrows and circle represent excitatory and inhibitory connections between nodes. See Materials and methods for details on model parameters. Postsynaptic potentials (PSPs) illustrate the internal dynamics

*Figure 2 continued on next page*

*Figure 2 continued*

of the gap detector (GAP, middle) and coincidence detector (COINC, bottom) in response to simulated VOT stimuli (line color). Outputs (panels F/G) are triggered by suprathreshold instantaneous PSPs ($\Sigma$PSP$\geq\theta$, dark lines) but not by subthreshold PSPs ($\Sigma$PSP$<\theta$; semitransparent lines). (F) Model outputs (a.u. = arbitrary units) evoked by simulated VOT stimuli for GAP (one cycle = 10 ms). Note that outputs for 0 ms and 10 ms VOTs are overlapping. No error bars shown because model simulations are deterministic. Grey box marks average peak window (across panels F/G); width matches peak window of real neural data (panels B/C). (G) Model outputs for COINC (H) Peak response to each simulated VOT stimulus for GAP (left) and COINC (right).

The online version of this article includes the following figure supplement(s) for figure 2:

**Figure supplement 1.** Connection weights between model nodes.

responses, overall, for voiceless stimuli (30–50 ms VOTs) compared to voiced stimuli (0–20 ms VOTs). Across V- electrodes, we found that voiceless stimuli with longer VOTs (i.e., closer to the preferred category's 50 ms endpoint VOT) also elicit increasingly stronger responses (Wilcoxon signed-rank test: $z = 3.52$, p=$4.4\times10^{-4}$). At the same V- sites, however, within-category VOT does not reliably predict response amplitude among (non-preferred) voiced stimuli (Wilcoxon signed-rank test: $z = -1.60$, p=0.11; *Figure 2B*: differences among solid blue lines but not dashed red lines). Across all V- and V+ electrodes, peak high-gamma response amplitude encoded stimulus VOT within the preferred category (Wilcoxon signed-rank test: $z = 6.02$, p=$1.7\times10^{-9}$), but not the non-preferred category (Wilcoxon signed-rank test: $z = 1.31$, p=0.19). While V- electrodes encoded sub-phonetic VOT more robustly within the voiceless category than within the voiced category (*Figure 2D*, left; Wilcoxon signed-rank test: $z = 3.00$, p=$2.7\times10^{-3}$), the opposite pattern emerged for V+ electrodes, which encoded sub-phonetic VOT more robustly within the voiced category than within the voiceless category (*Figure 2D*, right; Wilcoxon signed-rank test: $z = 3.78$, p=$1.6\times10^{-4}$).

Together, these analyses revealed two key results: (1) VOT encoding in human STG is not purely categorical, but also (2) the relationship between response amplitude and VOT is not linear across the entire continuum (*Figure 2D*). These results suggest that, even at the level of STG, the brain maintains information about the specific, sub-phonetic details of individual speech sounds. The asymmetrical pattern of within-category encoding suggests that individual neural populations in human auditory cortex encode information about both the category identity of a speech sound and its more fine-grained acoustic properties, or its *category goodness* (*Kuhl, 1991*; *Blumstein et al., 2005*; *Myers, 2007*).

## A simple neural network model of VOT encoding in STG

Thus far, we have demonstrated that a temporal cue that distinguishes speech sounds is represented by a spatial/amplitude code (*Ferster and Spruston, 1995*; *Shadlen and Newsome, 1994*) in human STG. To understand how this could be implemented computationally in the brain, we built an architecturally minimalistic neural network (*Figure 2E*, top). The network was designed to implement a small set of basic computations, motivated by well-established models of temporal processing (*Buonomano and Merzenich, 1995*; *Gao and Wehr, 2015*; *Eggermont, 2000*; *Carr, 1993*; *Konishi, 2003*; *Rauschecker, 2014*; *Rauschecker, 1998*). Specifically, our model employs discrete integrator units that detect temporal gaps or coincidences between distinct spectral events by incorporating canonical neurophysiological mechanisms that allow current input to modulate a unit's sensitivity to subsequent input in highly specific ways.

The entire model is comprised of just five localist units: a burst detector, a voicing detector, a gap detector (*G*AP), a coincidence detector (*C*OINC), and an inhibitory unit. Conventional leaky integrator dynamics governed continuously varying activation values of each rectified linear unit within the model (*McClelland and Rumelhart, 1981*; *McClelland et al., 2014*), with the activity $a_i(t)$ of a given unit $i$ at time $t$ depending on its prior activity $a_i(t-1)$, the weighted sum of its excitatory and inhibitory inputs $\sum_j w_{ji} * a_j(t-1)$, and unit-specific activation parameters (e.g., propagation threshold [$\theta$], decay rate). To illustrate intuitively how time-dependent neuronal properties can give rise to spatially-localized temporal cue processing, model parameters and connection weights were set manually (see Methods for details; *Figure 2—figure supplement 1*; *Supplementary file 2*). We presented the network with simplified inputs mimicking the spectral and temporal properties of the six VOT stimuli used in the ECoG experiment (*Figure 1A*; see Materials and methods; *Supplementary file 3*). Presentation of burst and voicing inputs triggered propagation of activation

that spread through the network, and our analyses assessed how the resulting activation dynamics differed depending on VOT.

The simulated responses of $G_{AP}$ and $C_{OINC}$ to VOTs of 0–50 ms are shown in *Figure 2F/G*. We observed striking qualitative similarities between $G_{AP}$ simulated outputs (*Figure 2F*) and the real neural responses of V- electrodes (*Figure 2B*), and between $C_{OINC}$ outputs (*Figure 2G*) and the V+ electrodes (*Figure 2C*). By design, voicing category is clearly distinguished in both $G_{AP}$ and $C_{OINC}$, with $G_{AP}$ responding more strongly to longer (voiceless) VOTs (30–50 ms), and $C_{OINC}$ responding more strongly to shorter (voiced) VOTs (0–20 ms). This demonstrates that spatial encoding of temporal cues (gaps vs. coincidences) can arise naturally within a simple, biologically-inspired neural network (*Buonomano and Merzenich, 1995*; *Gao and Wehr, 2015*; *Eggermont, 2000*; *Carr, 1993*; *Konishi, 2003*; *Rauschecker, 2014*; *Rauschecker, 1998*).

Perhaps more surprisingly, we also found that both $G_{AP}$ and $C_{OINC}$ detector units exhibit sensitivity to within-category VOT distinctions (*Figure 2H*). These partially graded activations mirror the pattern observed in the neural data (*Figure 2D*), where V- electrodes and $G_{AP}$ units are only sensitive to differences among long (voiceless) VOTs, and V+ electrodes and $C_{OINC}$ units are only sensitive to differences among short (voiced) VOTs.

These relatively sophisticated dynamics are the natural result of well-established computational and physiological mechanisms. Within the model, the burst and voicing detector units are tuned to respond independently to distinct spectral cues in the simulated acoustic input. Hence, the relative timing of their responses, but not their amplitudes, differ as a function of VOT. Both the gap ($G_{AP}$) and the coincidence ($C_{OINC}$) detector units receive excitatory input from both the burst and voicing detector units, but $G_{AP}$ and $C_{OINC}$. differ in how they integrate these inputs over time. Specifically, as described below, while initial excitatory input (from the burst detector) temporarily *decreases* the sensitivity of $G_{AP}$ to immediate subsequent excitatory input (from the voicing detector), the opposite is true of $C_{OINC}$.

In particular, prior work has shown that one computational implementation of gap detection involves configuration of a *slow inhibitory postsynaptic potential* (*IPSP*) microcircuit (*Figure 2E*, middle) (*Buonomano and Merzenich, 1995*; *Gao and Wehr, 2015*; *Douglas and Martin, 1991*; *McCormick, 1989*). In our model, activity in the burst detector following burst onset elicits fast suprathreshold *excitatory postsynaptic potentials* (*EPSPs*) in both $G_{AP}$ and the inhibitory unit, immediately followed by a longer-latency ('slow') IPSP in $G_{AP}$. This slow IPSP renders $G_{AP}$ temporarily insensitive to subsequent excitatory input from the voicing detector, meaning that voicing-induced excitation that arrives too soon (e.g., 10 ms) after the burst input, when inhibition is strongest, is not able to elicit a second suprathreshold EPSP in $G_{AP}$. Consequently, all short VOTs (below some threshold) elicit uniformly weak responses in $G_{AP}$ that reflect only the initial excitatory response to the burst (see, e.g., indistinguishable responses to 0 ms and 10 ms VOTs in *Figure 2F*). However, as $G_{AP}$ gradually recovers from the burst-induced slow IPSP, later-arriving voicing input (i.e., longer VOTs) tends to elicit suprathreshold responses that grow increasingly stronger with longer gaps, until $G_{AP}$ has reached its pre-IPSP (resting) baseline. In this way, our implementation of gap detection naturally captures three key patterns observed across V- electrodes (*Figure 2H*, left; *Figure 2D*, left): (1) amplitude encoding of a temporally cued category (selectivity for gaps over coincidences); (2) amplitude encoding of within-category differences in the preferred category (amplitude differences among gaps of different durations); and (3) no amplitude encoding of differences within the non-preferred category (uniformly lower amplitude responses to short VOTs of any duration).

In contrast, coincidence detection (*Carr, 1993*; *Konishi, 2003*; *Rauschecker, 2014*; *Margoliash and Fortune, 1992*; *Peña and Konishi, 2001*; *Pena and Konishi, 2002*; *Figure 2E*, bottom) emerges in the model because activity in the burst detector evokes only a subthreshold EPSP in $C_{OINC}$, temporarily increasing the sensitivity of $C_{OINC}$ to immediate subsequent excitatory input (from the voicing detector). During this period of heightened sensitivity, voicing-induced excitatory input that arrives simultaneously or after short lags can elicit larger amplitude (additive) EPSPs than could voicing-induced excitatory input alone. Because the magnitude of the initial burst-induced EPSP gradually wanes, the summation of EPSPs (from the burst and voicing) is greatest (and hence elicits the strongest response) for coincident burst and voicing (0 ms VOT), and the magnitude of the $C_{OINC}$ response to other voiced stimuli (e.g., 10–20 ms VOTs) becomes weaker as the lag between burst and voicing increases. Finally, in voiceless stimuli, since voicing arrives late enough after the burst (30+ ms) that there is no residual boost to the $C_{OINC}$ baseline post-synaptic potential,

elicited responses are entirely driven by a suprathreshold voicing-induced EPSP that reaches the same peak amplitude for all voiceless stimuli. Thus, our implementation of coincidence detection captures three key patterns observed in V+ electrodes (*Figure 2H*, right; *Figure 2D*, right): (1) amplitude encoding of a temporally cued category (selectivity for coincidences over gaps); (2) amplitude encoding of within-category differences in the preferred category (amplitude differences among stimuli with short VOTs); and (3) no amplitude encoding of differences within the non-preferred category (uniformly lower amplitude responses to long VOTs of any duration).

In summary, the neurophysiological dynamics underlying local STG encoding of VOT can be modeled using a simple, biologically-inspired neural network. The computational model captures both the between-category (phonetic) and within-category (sub-phonetic) properties of observed neural representations via well-established physiological mechanisms for gap and coincidence detection (*Buonomano and Merzenich, 1995*; *Gao and Wehr, 2015*; *Eggermont, 2000*; *Carr, 1993*; *Konishi, 2003*; *Rauschecker, 2014*; *Rauschecker, 1998*).

## Mechanisms that explain local category selectivity also predict early temporal dynamics

Thus far, we have focused on the encoding of speech sounds that differ in VOT based on activity patterns around the peak of the evoked response. However, in comparing the real and simulated neural data (*Figure 2*), we also observed a qualitative resemblance with respect to the onset latencies of evoked responses. Specifically, the timing of the evoked neural responses (relative to burst onset) appeared to depend on stimulus VOT in V+ electrodes and in the coincidence detector ($C_{OINC}$) unit (*Figure 2C/G*), but not in V- electrodes or in the gap detector ($G_{AP}$) unit (*Figure 2B/F*). This pattern could suggest that early temporal dynamics of the evoked response contribute to the pattern of category selectivity observed at the peak.

We examined the neural activity evoked by each VOT stimulus in V- and V+ electrodes at the onset of the response, typically beginning approximately 75–125 ms after stimulus (burst) onset. In the same two example electrodes from *Figure 1E*, we observed clear differences in the relationship between response onset latency and VOT (*Figure 3A*). To quantify the onset latency for each electrode to each VOT stimulus, we found the first timepoint after stimulus onset where the evoked high gamma response exceeded 50% of the electrode's overall peak amplitude (grand mean across

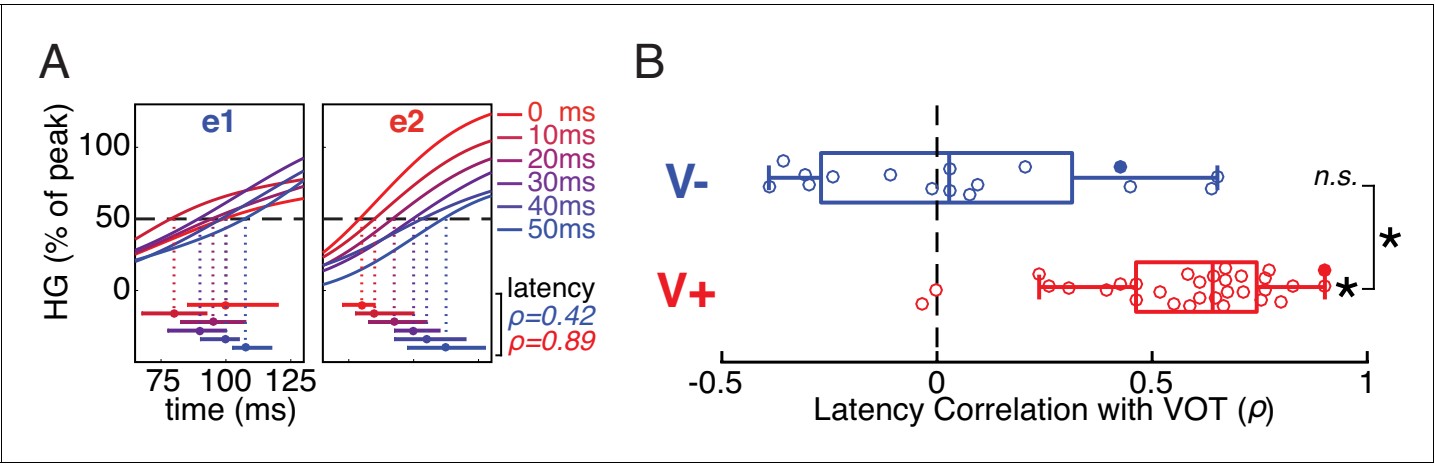

**Figure 3.** Early temporal dynamics of stimulus-evoked neural responses differ between voiceless-selective (V-) and voiced-selective (V+) electrodes. (**A**) Normalized trial-averaged HG responses to each VOT stimulus (line color) in two example electrodes (e1 and e2; same electrodes shown in *Figure 1D/E*). The time window (x-axis) is relative to onset of the burst and precedes the peak response. Horizontal bars show estimates (bootstrapped mean ± SE) of response onset latency for each VOT (first timepoint exceeding 50% of electrode's average peak HG). Mean bootstrapped rank-based correlation (Spearman's ρ) between VOT and response onset latency shown for e1 (blue) and e2 (red). (**B**) Across all V- electrodes, the bootstrapped correlation coefficients did not differ significantly from 0, suggesting that onset latency was time-locked to the burst. In contrast, across all V+ electrodes, the bootstrapped correlation coefficients were reliably positive (longer latencies for longer VOTs), and greater than for V- electrodes. Circles represent individual electrodes (filled: example electrodes in **A**). Boxes show interquartile range; whiskers extend to maximum/minimum of each group (excluding two outlier V+ electrodes); vertical bars are medians. Asterisks indicate significance ($p < 10^{-4}$; n.s. = not significant).

conditions). The rank correlation between VOT and response onset latency for e1 (a V- electrode) was substantially lower (Spearman's ρ = 0.42) than for e2 (a V+ electrode; ρ = 0.89).

A bootstrapped rank-based correlation coefficient was computed for each V- and V+ electrode (1000 resamples; see Methods). We found that response onset latency was strongly associated with VOT for V+, but not V-, electrodes (Wilcoxon signed-rank tests: V+, p=$1.6\times10^{-6}$; V-, p=0.57), and this difference between the two electrode types was highly reliable (Mann-Whitney rank-sum test: p=$1.7\times10^{-5}$) (*Figure 3B*).

The association between VOT and response latency also differed in $G_{AP}$ versus $C_{OINC}$ units in the model simulations (*Figure 2F/G*), with VOT-dependent response latencies emerging for $C_{OINC}$, but not $G_{AP}$. Closer examination of the model's internal dynamics reveals how the same time-dependent mechanisms that give rise to peak amplitude encoding of VOT are also responsible for these early temporal dynamics. As described above, the category selectivity of $G_{AP}$ (voiceless) and $C_{OINC}$ (voiced) results from how each unit's subsequent activity is modulated after detection of the burst. While the burst always elicits a fast suprathreshold response in $G_{AP}$ (irrespective of VOT), the $C_{OINC}$ response to the burst alone is subthreshold (*Figure 2E*, middle vs. bottom). Consequently, the initial $G_{AP}$ response is evoked by the burst of any VOT stimulus, so the response onset latency (when aligned to burst onset) does not depend on VOT (*Figure 2F*). Conversely, the earliest suprathreshold $C_{OINC}$ response is triggered by the onset of voicing, so the response onset latency (relative to burst onset) is later for longer VOTs (*Figure 2G*). Thus, the same well-established physiological mechanisms that give rise to peak amplitude encoding of temporally-cued voicing categories also predict the early temporal dynamics we observe in real neural data.

Finally, *Figure 3* shows that, unlike during the peak response window (150–250 ms after stimulus onset; *Figure 1F*), temporal information does encode VOT during an earlier window around the neural response onset in some neural populations. Indeed, both sub-phonetic and phonetic category-level information are carried by the onset latency of V+ electrodes, with evoked responses arising later at these sites for stimuli with progressively longer VOTs. Critically, the modeling results indicate that both the amplitude encoding patterns during the peak window and the temporal encoding patterns during the earlier onset window are captured by the same canonical neurophysiological mechanisms.

## Discussion

This study investigated how voice-onset time (VOT), a temporal cue in speech, is represented in human auditory cortex. Using direct intracranial recordings, we found discrete neural populations located primarily on the bilateral posterior and middle STG that respond preferentially to either voiced sounds, where the onset of voicing is coincident with the burst or follows it after a short lag (20 ms or less), or voiceless sounds, where the onset of voicing follows the burst after a temporal gap of at least 30–50 ms.

Past work has also found that phonetic information about speech sounds is encoded in the amplitude of evoked neural responses at spatially localized cortical sites (*Mesgarani et al., 2014*). In that work, however, STG activity was shown to encode the spectral properties of speech sounds most robustly, such as whether a phoneme is a vowel or a consonant and whether a consonant's spectrum is broadband (as in plosives, like /b/ and /p/) or is dominated by acoustic energy at high frequencies (as in fricatives, like /f/ and /s/).

The present results extend these earlier findings in a critical way, suggesting that the cortical representation of both spectral and temporal cues in speech follow a common spatial coding scheme. This result is also consistent with prior reports that neural response amplitude depends on VOT (*Mesgarani et al., 2014*), but such results have often involved natural speech stimuli where voicing categories varied along many other spectral acoustic dimensions besides the temporal cue (*Lisker, 1986*; *Soli, 1983*; *Stevens and Klatt, 1974*; *Summerfield and Haggard, 1977*). Here, the digitally synthesized VOT stimuli were tightly controlled to vary only in the relative timing of two invariant spectral cues (burst and voicing), thereby demonstrating that this temporal speech cue is encoded in the peak high-gamma response amplitude of spatially distinct neural populations in human STG.

While the present results clearly implicate a spatial/amplitude code in the cortical representation of VOT, other work has described VOT-dependent temporal response patterns that can also be

used to encode voicing categories (*Eggermont, 1995*; *Eggermont and Ponton, 2002*; *Liégeois-Chauvel et al., 1999*). For instance, Steinschneider and colleagues have observed neurons and neuronal populations in primate and human auditory cortices in which short VOTs elicit a single-peaked neural response, while longer VOTs elicit a double-peaked response (*Steinschneider et al., 1999*; *Steinschneider et al., 2013*; *Steinschneider et al., 2011*; *Steinschneider et al., 2005*; *Steinschneider et al., 1994*; *Steinschneider et al., 1995*; *Steinschneider et al., 2003*). Under this 'local' temporal coding model, the precise temporal dynamics of the response evoked at a single cortical site could distinguish voiced from voiceless VOTs. Our examination of the timing and amplitude of three peaks in the auditory evoked local field potentials of VOT-sensitive electrodes confirmed that such patterns do appear in some electrodes (*Figure 1—figure supplements 3* and *4*), clearly demonstrating that temporal and amplitude codes for VOT are not mutually exclusive (see also temporal encoding patterns in onset latencies of V+ electrodes; *Figure 3*). However, as with spectrally-defined phonetic contrasts (e.g., plosive vs. fricative; *Mesgarani et al., 2014*), it clear that the amplitude of the peak high-gamma (and, in many cases, of the LFP) response emerged as a robust representation of voicing category and of VOT.

VOT could also be encoded in the relative timing of responses in spatially-distributed, spectrally-tuned burst- and voicing-selective neural populations. Under this 'ensemble' temporal coding model (*Theunissen and Miller, 1995*; *Engineer et al., 2008*), the pattern of neural activity evoked by voiced VOTs (characterized by roughly coincident burst and voicing cues) would differ from the pattern evoked by voiceless VOTs in the precise temporal latency of the response in a vowel-selective neural population (a voicing detector) compared to the response in a plosive-selective neural population (a burst detector). However, the fact that we found cortical sites in every participant that exhibited robust category-dependent differences in their peak response amplitude rules out the possibility that at least these neural populations are merely responding to spectral cues in the burst or voicing alone.

Notably, if either (or both) of these models – a local or ensemble temporal code – was primarily responsible for the neural representation of VOT in the high-gamma range, then the selective corruption of temporal information in a classifier (*Figure 1F*) should have reduced neural decoding of voicing category to chance levels, while corrupting peak amplitude information should have had little or no effect. We found the opposite pattern of results: corrupting peak amplitude information had a devastating effect on the decoding of voicing category, while corrupting the fine temporal patterns that could have discriminated between voicing categories had no measurable impact on classifier performance. To be clear, our work does not rule out the possibility that local or ensemble temporal codes may also play a role in the cortical representation of VOT. However, it does highlight spatially-localized peak neural response amplitude as a robust code for VOT. Thus, in contrast to prior work theorizing parallel, but fundamentally different, coding schemes for spectrally- and temporally-cued phonetic features (*Steinschneider et al., 1999*; *Steinschneider et al., 2013*), we demonstrate evidence for a shared representation of both by high-gamma responses in the human superior temporal lobe.

In order to explicitly test potential computational and physiological mechanisms that could give rise to the observed spatial coding scheme, we implemented an architecturally simple neural network model. Although it is well known that spectral information is represented by a spatial neural code from the earliest stages of auditory transduction in the cochlea (*Eggermont, 2001*; *Oxenham, 2018*), the emergence of a spatial code for the representation of temporally-distributed cues in a transient acoustic signal poses a nontrivial computational problem. Our model highlights one parsimonious approach by which selectivity for either temporal gaps or coincidences could be implemented by biologically-inspired neurophysiological microcircuits (*Buonomano and Merzenich, 1995*; *Gao and Wehr, 2015*; *Eggermont, 2000*; *Carr, 1993*; *Konishi, 2003*; *Rauschecker, 2014*; *Rauschecker, 1998*).

We found that, just like in the neural data, gap and coincidence detector units responded to simulated voiced (/*b*/) and voiceless (/*p*/) stimuli with different response amplitudes. As such, we need not invoke any specialized temporal code to account for the representation of temporally cued phonetic features. Rather, our results provide evidence implicating a common neural coding scheme in the neural representation of behaviorally relevant speech features, whether they are embedded within the instantaneous spectrum or the fine temporal structure of the speech signal. Recent ECoG evidence suggests an even more expansive view of the fundamental role of spatial coding in cortical

speech representation (*Yi et al., 2019*) in which different neural populations also encode pitch (*Tang et al., 2017*) and key properties of the speech envelope such as onsets and auditory edges (*Hamilton et al., 2018*; *Oganian and Chang, 2019*).

Crucially, although the neural network was only designed to discriminate between categories (i. e., gaps vs. coincidences), we also observed graded amplitude differences in response to different VOTs (*Figure 2H*), but only in an electrode's preferred category. These within-category patterns emerged naturally from the same computational properties that allowed the network to capture basic between-category encoding: (1) the relative responsiveness of each temporal integrator unit ($G_{AP}$, $C_{OINC}$) to its various inputs (burst, voicing, and inhibition); (2) the time-dependent properties inherent to neuronal activation dynamics (e.g., decay of postsynaptic potentials towards a unit's resting activation level); and (3) the nonlinear transformation of postsynaptic inputs into response outputs (rectified linear activation function controlled by a unit's propagation threshold).

This asymmetric within-category encoding scheme closely resembled the pattern observed in real neurophysiological data, where peak response amplitude to VOTs within the same voicing category only differed within a neural population's preferred category (*Figure 2D*). This result clearly demonstrates that human nonprimary auditory cortex maintains a robust, graded representation of VOT that includes the sub-phonetic details about how a particular speech token was pronounced (*Blumstein et al., 2005*; *Toscano et al., 2010*; *Toscano et al., 2018*; *Frye et al., 2007*). Even though sub-phonetic information is not strictly necessary for mapping sound to meaning in stable, noise-free listening environments, this fine-grained acoustic detail has demonstrable effects on listeners' behavior (*Kuhl, 1991*; *Carney, 1977*; *Pisoni and Tash, 1974*; *Massaro and Cohen, 1983*; *Andruski et al., 1994*; *McMurray et al., 2002*; *Schouten et al., 2003*), and modern theories of speech perception agree that perceptual learning (e.g., adaptation to accented speakers) and robust cue integration would be impossible if the perception of speech sounds were strictly categorical (*Miller and Volaitis, 1989*; *Clayards et al., 2008*; *Kleinschmidt and Jaeger, 2015*; *McMurray and Jongman, 2011*; *Toscano and McMurray, 2010*; *McClelland and Elman, 1986*; *Norris and McQueen, 2008*; *Norris et al., 2016*; *Magnuson et al., 2020*). Crucially, these data suggest that the same spatial/amplitude code that is implicated in the representation of *phonetic* information (from spectral or temporal cues) can also accommodate the representation of *sub-phonetic* information in the speech signal.

The onset latency results (*Figure 3*) established an entirely novel correspondence between the real and simulated results that extended beyond the peak response window. Response onset latencies of V- electrodes were time-locked to the burst (*Figures 2B* and *3*), while responses of V+ electrodes were time-locked to voicing onset (*Figures 2C* and *3*). These highly reliable neurophysiological results neatly match specific predictions of our parsimonious model without the need to postulate additional mechanisms (*Figure 2F/G*).

The correspondence between simulated and real neural data in the onset latency results may also have implications for the question of whether the observed temporal integration is occurring locally in STG or is inherited from earlier levels of auditory processing (e.g., from midbrain or primary auditory cortex). The model's gap and coincidence detectors ($G_{AP}$, $C_{OINC}$) are designed to directly simulate neural populations in the STG. Their inputs from the burst and voicing detectors are only spectrally processed, so in the model, the temporal onset latency dynamics (*Figure 2F/G*) first arise in $G_{AP}$ and $C_{OINC}$. As such, the fact that the model's prediction is borne out in the neural data in STG (*Figure 2B, C* and *3*) is consistent with local temporal integration in STG. While these modeling results do not definitively rule out temporal integration at lower levels of the ascending auditory pathway, its potentially local emergence in high-order auditory cortex illustrates how even relatively simple computational models can be used to generate novel hypotheses, which can ultimately be tested in real neurophysiological data.

Overall, the results of these model simulations illustrate how the same network properties that transform temporal cues into a spatial code are also able to naturally explain at least three additional patterns observed within category-selective neural populations: (1) the graded encoding of VOT within a population's preferred category; (2) the lack of graded encoding of VOT within a population's non-preferred category; and (3) the early temporal dynamics of neural responses, which depend on a population's category-selectivity. Thus, the model provides an explicit, mathematical account of multiple seemingly disparate observations about the neurophysiological data, all of which

arise directly from a parsimonious implementation of gap- and coincidence-detection with well-established, theoretically-motivated neuronal circuits.

The model we present is just one of many possible architectures that could capture these interesting properties of the neural response. For example, mechanisms like temporal delay lines (*Carr, 1993*; *Rauschecker, 2014*) could also be used to implement gap detection. Broadly, we chose to implement a simple hand-tuned neural network model to maximize our ability to explore the detailed dynamics we observed in the neural data. Our approach follows a rich history of using these types of hand-tuned models to explain a wide array of cognitive and perceptual phenomena (including the perception of VOT in speech), as exemplified by the influential TRACE model of speech perception (*McClelland and Elman, 1986*). An alternative approach to modeling VOT perception is to train a neural network to distinguish voiced from voiceless sounds based on distributed activation dynamics within biologically-grounded spectral processing maps (*Damper, 1994*). Our model borrows aspects of these two approaches (hand-tuning; biological plausibility) and it extends this past work by directly modeling the time-dependent mechanisms that could give rise to continuously-varying neural responses in STG.

While the model captured several notable features of the neural data (including some for which it was not explicitly designed), we observed at least one inconsistency between the simulated and real neural responses. The model predicted VOT-dependence in the latency of the *peak* response in both $G_{AP}$ and $C_{OINC}$ units (*Figure 2F/G*), but we did not find evidence for these fine-grained patterns in the high-gamma data (*Figure 2B/C*; see also lack of category-dependent temporal patterns during peak window: *Figure 1F*). However, it is unclear whether this is a false prediction of the model, or whether we did not observe the effect in the neural data because of, for example, poor signal-to-noise ratio for this effect. Regardless of whether the discrepancy arises from the model or the real data, it represents a gap in our mechanistic understanding of the processing of this phenomenon, and should therefore be a target for further research.

Although topographic functional organization is pervasive among many spatial neural coding schemes described in sensory neuroscience, including for the representation of spectral and temporal acoustic cues in audition (e.g., tonotopy in mammalian auditory cortex; *Eggermont, 2001*; *Oxenham, 2018* or chronotopy in bats; *Kössl et al., 2014*; *Portfors and Wenstrup, 2001*), this functional organization seems not to extend to the spatial code for speech on the lateral temporal cortex in humans. As with tuning for spectrally-cued phonetic features (*Mesgarani et al., 2014*; *Hamilton et al., 2018*) (e.g., plosives vs. fricatives), VOT-sensitive neural populations in the present study were scattered throughout posterior and middle superior temporal gyrus with no discernible topographical map of selectivity or evidence for lateralized asymmetries (*Liégeois-Chauvel et al., 1999*; *Zatorre and Belin, 2001*), although data limitations prevent us from ruling out this possibility entirely (for detailed results, see Material and methods).

Most of the present analyses focused on the high-gamma component of the neural response, but this work does not discount a potential role for lower-frequency oscillations in speech perception (*Fries, 2009*; *Giraud and Poeppel, 2012*) or in the perception of phonemes (*Kösem et al., 2018*; *Peelle and Davis, 2012*). Indeed, it is clear from the exploratory analyses of auditory evoked local field potentials (*Figure 1—figure supplements 3* and *4*) that there do exist complex associations between VOT and the amplitude/temporal information carried in lower-frequency ranges. Future work should systematically investigate the relationship between high-gamma and other neural signals (such as the local field potential), their relative contributions to the perceptual experience of and neural representation of speech, and the importance of detailed temporal information in each (see, e.g., *Nourski et al., 2015*).

Finally, it is critical to distinguish our results from studies describing neural correlates of categorical speech perception, per se (e.g., *Chang et al., 2010*). Neural responses to different VOT tokens that are members of the same voicing category can only be considered truly categorical if the responses are indiscriminable (e.g., *Liberman et al., 1957*; *Macmillan et al., 1977*). In our results, acoustically distinct members of the same phonetic category <u>are</u> distinguishable in neural populations that are selective for that voicing category (*Figure 2*). In light of this graded VOT representation, the present results are best interpreted as elucidating neural mechanisms of category perception, but not necessarily categorical perception, of voiced vs. voiceless stop consonants. While limited coverage beyond the superior temporal lobe precludes us from ruling out the influence of top-down categorical perception (*Lee et al., 2012*; *Myers et al., 2009*; *Evans and Davis,*

*2015*) (possibly originating in frontal cortex; *Sohoglu et al., 2012*; *Leonard et al., 2016*; *Cope et al., 2017*; *Park et al., 2015*) on our results, it is notable that the model we present (which does not posit top-down effects) suggests that top-down effects may not be a necessary condition for explaining the observed non-linear encoding patterns (see also *McClelland et al., 2006*; *McQueen et al., 2006*; *Norris et al., 2000*; *McClelland and Elman, 1986*; *Norris and McQueen, 2008*).

In conclusion, the present results show that spatially-discrete neural populations in human auditory cortex are tuned to detect either gaps or coincidences between spectral cues, and these sites simultaneously represent both phonetic and sub-phonetic information carried by VOT, a temporal speech cue found in almost all languages (*Lisker and Abramson, 1964*; *Cho and Ladefoged, 1999*). This demonstrates a common (spatial) neural code in STG that accounts for the representation of behaviorally relevant phonetic features embedded within the spectral and temporal structure of speech. From a simple model that transforms a temporal cue into a spatial code, we observed complex dynamics that show how a highly variable, continuous sensory signal can give rise to partially abstract, discrete representations. In this way, our findings also add to a growing body of work highlighting the critical role of human STG as a sensory-perceptual computational hub in the human speech perception system (*Yi et al., 2019*; *Tang et al., 2017*; *Chang et al., 2010*; *Leonard et al., 2016*; *DeWitt and Rauschecker, 2012*; *Obleser and Eisner, 2009*; *Leonard and Chang, 2014*; *Sjerps et al., 2019*).

## Materials and methods

### Data and code availability

All data and code associated with this study and necessary for replication of its results are available under a Creative Commons license at the associated Open Science Framework project page (https://osf.io/9y7uh/) (*Fox et al., 2020*).

### Participants

A total of seven human participants with self-reported normal hearing were implanted with high-density (128 or 256 electrodes; 4 mm pitch) multi-electrode cortical ECoG surface arrays as part of their clinical treatment for epilepsy. Placement of electrode arrays was determined based strictly on clinical criteria. For all patients who participated in this study, coverage included peri-Sylvian regions of the lateral left (n = 3) or right (n = 4) hemisphere, including the superior temporal gyrus (STG). All participants gave their written informed consent before the surgery and affirmed it at the start of each recording session. The study protocol was approved by the University of California San Francisco Committee on Human Research. Data from two additional participants were excluded from analyses because of excessive epileptiform activity (artifacts) during recording sessions.

### Imaging

Electrode positions (*Figure 1D* and *Figure 1—figure supplement 2*) were determined from post-surgical computed tomography (CT) scans and manually co-registered with the patient's MRI. Details of electrode localization and warping to a standardized brain (MNI; *Figure 2A*) are described elsewhere (*Hamilton et al., 2017*).

### Stimuli

Stimuli (*Figure 1B*) were generated with a parallel/cascade Klatt-synthesizer KLSYN88a using a 20 kHz sampling frequency (5 ms frame width in parameter tracks). All stimulus parameters were identical across stimuli, with the exception of the time at which the amplitude of voicing began to increase (in 10 ms steps from 0 ms to 50 ms after burst onset; *Figure 1A*). The total duration of each stimulus was 300 ms regardless of VOT. The onset noise-burst was 2 ms in duration and had constant spectral properties across all stimuli. The dominant frequency ranges for the vowel were: F0 = 100 Hz; F1 = 736 Hz; F2 = 1221 Hz; F3 = 3241 Hz (consistent with a vocal tract length of 13.5 cm). Formant transitions always began at 30 ms. The vowel's amplitude began ramping down 250 ms after stimulus onset. The stimuli are made available among this study's supplementary materials and at the associated Open Science Framework page (*Fox et al., 2020*).

## Behavioral procedure

During ECoG recording, the VOT stimuli were presented monaurally over free-field loudspeakers at a comfortably listening level via a custom MATLAB script (*Fox et al., 2020*) in a blocked pseudorandom order. Four of seven participants simultaneously performed a behavioral task wherein they indicated on each trial whether they heard 'ba' or 'pa' using a touchscreen tablet (programmed using a custom MATLAB GUI). In these recording sessions, the onset of the next trial began 500 ms after a response was registered or 5 s after the end of the stimulus (if no response was registered). In sessions where participants chose to listen to the stimuli passively (instead of participating in the behavioral task), the onset of the next trial began approximately 1000 ms after the end of the previous trial. *Supplementary file 1* reports number of trials per participant.

## Behavioral analysis

For the four participants who participated in the behavioral identification task, individual trials were excluded from behavioral analysis if a participant did not make a response or if the participant's reaction time was more than three standard deviations from the participant's mean reaction time.

Behavioral response data were submitted to mixed effects logistic regression with a fixed effect of VOT (coded as a continuous variable) and random intercepts for participants, allowing individual participants to vary in their voicing category boundary. Using the best-fit model estimates, we calculated the overall voicing category boundary across all participants ($\chi$ = 21.0ms; *Figure 1—figure supplement 1*, panel A) and in the each individual participant (after adjusting for random intercept fit for each participant; *Figure 1—figure supplement 1*, panel B, and *Figure 1C*) as follows (*Feldman et al., 2009*), where $\beta_0$ is the best-fit intercept and $\beta_{VOT}$ is the best-fit effect of slope:

$$\chi = -\frac{\beta_0}{\beta_{VOT}}$$

## ECoG signal processing

### Recording and preprocessing

Voltage fluctuations were recorded and amplified with a multichannel amplifier optically connected to a digital signal acquisition system (Tucker-Davis Technologies) sampling at approximately 3051.78 Hz. Line noise was removed via notch filtering (60 Hz and harmonics at 120 and 180 Hz) and the resulting time series for each session was visually inspected to exclude channels with excessive noise. Additionally, time segments with epileptiform activity were excluded. The time series data were then common-average referenced (CAR) to included electrodes either across an electrode's row in a $16 \times 16$ channel grid or across the entire grid depending on the technical specifications of the amplifier used for a given participant.

### High-gamma extraction

The analytic amplitude of the high-gamma (HG; 70–150 Hz) frequency band was extracted by averaging across eight logarithmically-spaced bands with the Hilbert transform as described elsewhere (*Mesgarani et al., 2014*; *Sjerps et al., 2019*). The HG signal was down-sampled to 400 Hz, providing temporal resolution to observe latency effects on the order of <10 ms (the spacing of the VOTs of among the six experimental stimuli).

### Trial alignment and extraction

Trial epochs were defined as 500 ms before to 1000 ms after each stimulus onset. Trials were excluded for all channels if the epoch window contained any time segments that had been marked for exclusion during artifact rejection. The HG signal for each trial was z-scored based on the mean and standard deviation of a baseline window from 500 ms to 200 ms before stimulus onset. A 50 ms moving average boxcar filter was applied to the HG time series for each trial.

### Local field potential extraction

Data for analyses of auditory evoked local field potentials consisted of the same raw voltage fluctuations (local field potential), preprocessed with identical notch filtering, CAR, artifact/channel

rejection, and down-sampling (to 400 Hz). Trial epochs (500 ms before to 1000 ms after each stimulus onset) were not z-scored.

## Electrode selection
### Speech-responsive electrodes
An electrode was included in our analyses if (1) it was anatomically located on the lateral temporal lobe (either superior or middle temporal gyrus), and (2) the electrode's grand mean HG (across all trials and timepoints during a window 100–300 ms after stimulus onset) exceeded one standard deviation of the baseline window's HG activity. Across all seven participants, 346 electrodes met these criteria (*speech-responsive electrodes*; *Supplementary file 1*; *Figure 1—figure supplement 2*).

### Peak neural response
The timepoint at which each speech-responsive electrode reached its maximum HG amplitude (averaged across all trials, irrespective of condition) was identified as that electrode's peak, which was used in the subsequent peak encoding analyses. Because we were focused on auditory-evoked activity in the temporal lobe, the search for an electrode's peak was constrained between 0 and 500 ms after stimulus onset. Electrode size in *Figure 1D* and *Figure 1—figure supplement 2* corresponds to this peak HG amplitude for each speech-responsive electrode.

### VOT-sensitive electrodes
To identify electrodes where the peak response depended on stimulus VOT (*VOT-sensitive electrodes*), we computed the nonparametric correlation coefficient (Spearman's ρ) across trials between VOT and peak HG amplitude. Because nonparametric (rank-based) correlation analysis measures the monotonicity of the relationship between two variables, it represents an unbiased ('model-free') indicator of amplitude-based VOT encoding, whether the underlying monotonic relationship is categorical, linear, or follows some other monotonic function (*Bishara and Hittner, 2012*). This procedure identified 49 VOT-sensitive electrodes across all seven participants (p<0.05; *Figure 2A* and *Figure 1—figure supplement 2*; *Supplementary file 1*). Electrode color in *Figure 1D* and *Figure 1—figure supplement 2* corresponds to the correlation coefficient at each electrode's peak (min/max ρ = ±0.35), thresholded such that all speech-responsive electrodes with non-significant (p>0.05) correlation coefficients appear as white.

This set of VOT-sensitive sites was then divided into two sub-populations based on the sign of each electrode's correlation coefficient (ρ): voiced-selective (V+) electrodes (n = 33) had significant ρ<0, indicating that shorter (more /b/-like; voiced) VOTs elicited stronger peak HG responses; voiceless-selective (V-) electrodes (n = 16) had significant ρ>0, indicating that longer (more /p/-like; voiceless) VOTs elicited stronger peak HG responses.

Across VOT-sensitive electrodes, the mean peak occurred 198.8 ms after stimulus onset (SD = 42.3 ms). The semi-transparent grey boxes in *Figures 1E* and *2B/C* illustrate this peak window (mean peak ± 1 SD).

## Analysis of VOT-sensitive electrodes
### Encoding of voicing category
Electrodes that exhibit a monotonic relationship between VOT and peak HG amplitude should also be likely to exhibit a categorial distinction between shorter (voiced) and longer (voiceless) VOTs. We conducted two analyses that confirmed this expectation. In each analysis, we computed a nonparametric test statistic describing the discriminability of responses to voiced vs. voiceless stimuli at each electrode's peak (z-statistic of Mann-Whitney rank-sum test) and then tested whether the population of test statistics for each group of electrodes (V- and V+) differed reliably from zero (Wilcoxon signed-rank tests). In the first analysis, voicing category was defined based on the psychophysically determined category boundary (voiced: 0–20 ms VOTs; voiceless: 30–50 ms VOTs), which allowed us to include all VOT-sensitive electrodes (n = 49) in the analysis, including electrodes from participants who did not complete the behavioral task (3/7 participants).

In the second analysis, a trial's voicing category was determined based on the actual behavioral response recorded for each trial (irrespective of VOT), so this analysis was not dependent on the assumption that the VOT continuum can be divided into two categories based on the average

boundary calculated across participants. This analysis examined the subset of trials with behavioral responses and the subset of VOT-sensitive electrodes found in the four participants with behavioral data (n = 27; 12 V- electrodes, 15 V+ electrodes) (*Supplementary file 1*).

Given the strong correspondence between the categorically defined VOT stimulus ranges (0–20 ms vs. 30–50 ms VOTs) and identification behavior (e.g., *Figure 1C*), the agreement between these results was expected.

Significance bars for the two example STG electrodes in one participant (e1 and e2; *Figure 1E*) we computed to illustrate the temporal dynamics of category selectivity. In these electrodes, we conducted the test of between-category encoding (Mann-Whitney rank-sum test; first analysis) at every timepoint during the trial epoch (in addition to the electrodes' peaks). Bars plotted for each electrode in *Figure 1E* begin at the first timepoint after stimulus onset where the significance level reached p<0.005 and ends at the first point thereafter where significance fails to reach that threshold (e1: 140 to 685 ms post onset; e2: 65 to 660 ms post onset).

## Encoding of VOT within voicing categories

Because VOT-sensitive electrodes were identified via nonparametric correlation analysis (Spearman's ρ) across all VOTs, the monotonic relationship between VOT and peak HG amplitude at these sites could be driven by the observed phonetic (between-category) encoding of voicing without any robust sub-phonetic (within-category) encoding of VOT. To assess sub-phonetic encoding of VOT in the peak response amplitude of VOT-sensitive electrodes, we computed the rank-based correlation (Spearman's ρ) between VOT and HG amplitude at each electrode's peak separately for trials in each voicing category (0–20 ms vs. 30–50 ms VOTs). The statistical reliability of within-category encoding was summarized by computing a test-statistic ($t$) for every correlation coefficient ($\rho_{0-20}$ and $\rho_{30-50}$ for each VOT-sensitive electrode) as follows:

$$t = \frac{\rho\sqrt{n-2}}{\sqrt{1-\rho^2}}$$

where $n$ is the number of trials with VOTs in a given voicing category. The resulting set of test statistics (one per voicing category per VOT-sensitive electrode) served as the basis for the following analyses of peak within-category encoding.

For each group of electrodes (V- and V+), we tested whether the encoding of VOT within each voicing category differed reliably from 0 (Wilcoxon signed-rank tests). We also conducted a Wilcoxon signed-rank test for each electrode group that compared the within-category correlation $t$-statistics for voiceless and voiced categories.

The above tests addressed the encoding properties of one electrode group at a time (either V- or V+ electrodes). Finally, a pair of Wilcoxon signed-rank tests combined across the full set of VOT-sensitive electrodes (n = 49) to summarize the within-category VOT encoding results within electrodes' (1) preferred and (2) non-preferred categories. In order to conduct this 'omnibus' test, we multiplied the correlation $t$-statistics for all V+ electrodes (for tests within each category) by −1. This simple transformation had the consequence of ensuring that positive correlation statistics always indicate stronger peak HG responses to VOTs that were closer to the endpoint of an electrode's preferred category.

## Visualizations of within-category VOT encoding

To visualize the pattern of within-category encoding of VOT in the peak HG amplitude of V- and V+ electrodes, we computed a normalized measure of the peak response amplitude to each VOT stimulus for each VOT-sensitive electrode. *Figure 2B and C* show the full time series of the average (± SE) evoked responses of V- and V+ electrodes to all six VOT stimuli. To show encoding patterns across electrodes with different peak amplitudes, each electrode's activity was normalized by its peak HG (grand mean across all VOTs). *Figure 2D* shows the amplitude of the average response evoked by a given VOT at a given electrode's peak relative to the average response evoked by the other VOT stimuli, or *peak HG (% of max)*, averaged across electrodes in each group (V-, left; V+, right) and participants (± SE). For each electrode, the mean HG amplitude evoked by each VOT at the peak was scaled and normalized by subtracting the minimum across all VOTs and dividing by the maximum across all VOTs after scaling.

## Neural response latency

The normalized HG responses used for *Figure 2B/C* were also used for the analysis of onset latency effects (*Figure 3*): *HG (normalized)* (*Figure 2B/C*) and *HG (% of peak)* (*Figure 3A*) are computationally equivalent. Neural response onset latency for an electrode was defined as the first timepoint at which its average response to a given VOT stimulus exceeded 50% of its peak HG (based on the peak of the grand average response across all VOTs). A bootstrapping with resampling procedure was employed to estimate the onset latencies of responses to different VOTs at each electrode and to assess any possible relationship between onset latency and VOT. During each sampling step in this procedure (1000 bootstrap samples), we computed the average time series of the normalized HG response to each VOT, the onset latency for the response to each VOT, and the nonparametric correlation (Spearman's $\rho$) between onset latency and VOT. Wilcoxon signed-rank tests asked whether the population of bootstrapped correlation coefficient estimates for each electrode group reliably differed from zero. A Mann-Whitney rank-sum test compared the VOT-dependency of response onset latency between electrode groups. Color-coded horizontal bars below the neural data in *Figure 3A* show onset latency estimates (mean ± bootstrap standard error) for responses to each VOT at two example electrodes. All electrodes were included in the analyses, but the bootstrapped correlation coefficient estimates for two V+ electrodes that were outliers (>3 SDs from median) were excluded from the visualized range of the box-plot's whiskers in *Figure 3B*.

## Population-based neural classification

For each participant, we trained a set of multivariate pattern classifiers (linear discriminant analysis with leave-one-out cross validation) to predict trial-by-trial voicing category (/*b*/: 0–20 ms VOTs vs. /*p*/: 30–50 ms VOTs) using HG activity across all speech-responsive electrodes on the temporal lobe during a time window around the peak neural response. The peak window was defined as beginning 150 ms and ending 250 ms after stimulus onset, selected based on the average and standard deviation of the peaks across all VOT-sensitive electrodes. We created four separate classifiers for each participant that allowed us to evaluate the contribution of amplitude and temporal structure to voicing category encoding (*Figure 1F*).

To corrupt the reliability of any spatially-localized amplitude information about whether the VOT stimulus presented to a participant on a given trial was a /*b*/ or a /*p*/, the neural responses at every electrode on every trial were normalized so that the average response to a /*b*/ and the average response to a /*p*/reached the same amplitude at each electrode's peak. Specifically, for each electrode, we found its peak (timepoint where the grand average HG time series across all trials reached its maximum), calculated the mean HG amplitude across all trials for VOTs within each category at that peak, and divided the HG values for every timepoint in a trial's time series by the peak HG amplitude for that trial's category. This amplitude normalization procedure forces the average amplitude of the neural response across all trials of /*b*/ and of /*p*/ to be equal at each electrode's peak, while still allowing for variation in the amplitude of any individual trial at the peak.

To corrupt the reliability of any timing information during the peak response window about whether the VOT stimulus presented to a participant on a given trial was a /*b*/ or a /*p*/, the timing of the neural response on every trial (across all electrodes) was randomly shifted in time so that the trial could begin up to 50 ms before or after the true start of the trial. Specifically, for each trial, a jitter value was drawn from a discrete (integer) uniform random distribution ranging between −20 to 20 (inclusive range) ECoG time samples (at 400 Hz, this corresponds to ±50 ms, with a mean jitter of 0 ms), and the HG time series for all electrodes on that trial was moved backward or forward in time by the number of samples dictated by the trial's jitter value. This temporal jittering procedure has the effect of changing whether the peak response window for a given trial is actually drawn from 100 to 200 ms after stimulus onset, 200–300 ms after stimulus onset, or some other window in between.

Crucially, this procedure will misalign any reliable, category-dependent differences in peak timing or temporal dynamics within individual electrodes or temporal patterns or relationships that exist across distributed electrodes. For instance, the peak window overlaps with a window during which past work examining intracranial auditory evoked local field potentials found evidence of waveform shape differences between responses of single electrodes to voiced and voiceless stimuli (single- vs. double-peaked responses; see, e.g., Figure 10 of *Steinschneider et al., 2011*). If similar temporal

differences in waveform shape existed in the present high-gamma data, the temporal jittering procedure would detect a contribution of temporal information to decoding. Moreover, to the extent that the peak of a trial's evoked high-gamma response occurs during or close to the peak window (either within one electrode ['local' temporal code] or across multiple electrodes in the same participant ['ensemble' temporal code]), the temporal jittering procedure would disrupt the reliability of this information to reveal the contribution of peak latency information to decoding accuracy. On the other hand, if the peak responses to stimuli from distinct voicing categories differ in the amplitude of the HG response at VOT-sensitive cortical sites, and if these differences persist throughout much of the peak window, then this temporal jittering procedure is unlikely to prevent the classifier from learning such differences.

For each participant, we trained one classifier where neither amplitude nor timing information were corrupted (+Amplitude/+Timing), one where only timing information was corrupted (+Amplitude/-Timing), one where only amplitude information was corrupted (-Amplitude/+Timing), and one where both were corrupted (-Amplitude/-Timing; here, amplitude normalization preceded temporal jittering). With each of these datasets, we then performed dimensionality reduction to minimize overfitting using spatiotemporal principal component analysis on the ECoG data for every electrode and all timepoints within the peak window (retaining PCs accounting for 90% of the variance across trials of all VOTs). Finally, training and testing of the linear discriminant analysis classifiers were conducted iteratively, holding out a single trial, training a classifier to predict voicing category using all other trials, and then predicting the voicing category of the held-out trial. For each participant and for each classifier, accuracy was the proportion of held-out trials that were correctly labeled. Wilcoxon signed-rank tests assessed and compared accuracy levels (across participants) achieved by the different models.

## Computational neural network model

### Overview of architecture and dynamics

A simple five-node, localist neural network (*Figure 2E*) was hand-connected to illustrate how time-dependent properties of neuronal units and their interactions can transform a temporal cue into a spatial code (responses of different amplitudes to different VOTs at distinct model nodes). A gap detector received excitatory input from both a burst detector and voicing detector, as well as input from an inhibitory node that only received excitatory input from the burst detector. This represented an implementation of a slow inhibitory postsynaptic potential (slow IPSP) circuit (*Buonomano and Merzenich, 1995*; *Gao and Wehr, 2015*; *Douglas and Martin, 1991*; *McCormick, 1989*). A coincidence detector received excitatory input from the burst and voicing detectors.

### Network connectivity

Weights between units in this sparsely connected, feedforward network were set according to a minimalist approach. All excitatory connections from the burst detector (to the inhibitory node, the gap detector, and the coincidence detector) had identical weights. All excitatory connections from the voicing detector (to the gap detector and the coincidence detector) had identical weights (stronger than from burst detector). *Figure 2—figure supplement 1* indicates all nonzero connection weights between the network's nodes, as illustrated in *Figure 2E*.

### Leaky-integrator dynamics

At the start of the model simulations, prior to the onset of any stimulus ($t = 1$), the activation level $a_i(t)$ of each node $i$ was set to its resting level ($\rho_i$). Simulations ran for 100 cycles, with 1 cycle corresponding to 10ms. On each subsequent cycle ($t \in [2, 100]$), activation levels of every node in the model were updated iteratively in two steps, as described in the following algorithm:

1. Decay: For every node $i$ with prior activation level $a_i(t-1)$ that differs from $\rho_i$, $a_i(t)$ decays towards $\rho_i$ by its decay rate ($\lambda_i$) without overshooting $\rho_i$.
2. Sum Inputs: For every node $i$, the total excitatory and inhibitory inputs are summed. This includes both model-external (clamped) inputs (i.e., from stimuli presented to the model) on the current cycle $t$ and model-internal inputs from other nodes based on their activation level on the prior cycle $a_j(t-1)$. Inputs from a presynaptic node $j$ can only affect the postsynaptic node $i$ if its prior activation $a_j(t-1)$ exceeds the presynaptic node's propagation threshold ($\theta_j$).

Summation of model-internal inputs within $i$ is weighted by the connection weights from the various presynaptic nodes (*Figure 2—figure supplement 1*): $\sum_{j} w_{ji} * a_j(t-1)$. The new activation level $a_i(t)$ is bounded by the node's minimum ($m_i$) and maximum ($M_i$) activation levels, irrespective of the magnitude of the net effect of the inputs to a node.

All activation parameters for all nodes are listed in *Supplementary file 2*. Minimum, maximum, and resting activation levels were identical across all units. Decay rates and propagation thresholds were identical across the burst and voicing detectors and the inhibitory node. The integrator units (gap and coincidence detectors) decayed more slowly than the other units, which could only affect other model nodes during one cycle. Activation levels in the coincidence detector had to reach a higher level (propagation threshold) to produce model outputs than in the gap detector, a difference which allowed the gap detector to register the fast suprathreshold response characteristic of slow IPSP circuits and allowed the coincidence detector to register a coincidence only when both burst and voicing were detected simultaneously or at a short lag.

## Model inputs

Two inputs were clamped onto the model in each simulation, representing the onset of the burst and of voicing (*Figure 1A*). The voicing input was only clamped onto the voicing detector at the onset of voicing. *Supplementary file 3* illustrates vectors describing each of the simulated VOT inputs.

## Sensitivity of model dynamics to variations in hand-tuned model parameters

Although most of the parameters of the model are theoretically uninteresting and were set to default levels (see *Supplementary file 2*), analysis of parameter robustness for the model revealed four primary sensitivities based on the relative values set for certain specific parameters. (1) and (2) below involve the propagation thresholds [$\theta$] of the temporal integrator units (**G**AP, **C**OINC), which allow the model to achieve gap and coincidence detection. (3) and (4) below involve the rate of decay of activation [$\lambda$] of the temporal integrator units, which dictate where along the VOT continuum the boundary between voicing categories lies.

1. Propagation threshold [$\theta$] of coincidence detector unit (**C**OINC): In our model, coincidence detection is achieved by preventing the coincidence detector (**C**OINC) from propagating an output in response to the burst until the voicing has arrived (hence responding with a higher-than-minimum peak amplitude only when the voicing is coincident with or arrives shortly after the burst). Thus, the propagation threshold for **C**OINC ($\theta_{\mathbf{Coinc.}}$) must be <u>greater than</u> the connection weight from the burst-detector to (**C**OINC). ($W_{Burst \rightarrow Coinc.}$).
2. Propagation threshold [$\theta$] of gap detector unit (**G**AP): On the other hand, the propagation threshold for the gap detector [**G**AP] ($\theta_{Gap}$) must be <u>less than</u> the connection weight from the burst-detector to **G**AP ($W_{BURST \rightarrow GAP}$) to register the fast suprathreshold response characteristic of slow IPSP circuits.

The primary factor affecting the location of the boundary between voiced (short VOTs) and voiceless (long VOTs) categories is the time-dependent rate of decay of postsynaptic potentials in **G**AP and **C**OINC towards the unit's resting activation level.

1. Rate of decay of activation [$\lambda$] in **C**OINC in comparison to connection weights from inputs to **C**OINC: For **C**OINC, the boundary is the VOT value after which there is no longer any additional boost to its peak amplitude from the initial burst, and this requires the decay rate of **C**OINC ($\lambda_{Coinc.}$) and the connection weight from the burst-detector to **C**OINC ($W_{BURST \rightarrow COINC}$) to be in balance. Increasing $\lambda_{Coinc.}$ or decreasing $W_{Burst \rightarrow Coinc}$ (independently) will move the boundary earlier in time.
2. Rate of decay of activation [$\lambda$] in **G**AP in comparison to connection weights from inputs to **G**AP: Similarly, for **G**AP, the category boundary is the VOT value before which the remaining influence of the initial inhibition is still so strong that the arrival of voicing input cannot exceed $\theta_{Gap}$. Increasing $\lambda_{Gap}$, decreasing $w_{Inhib. \rightarrow Gap}$, or increasing $w_{Voicing \rightarrow Gap}$ (independently) would each move the boundary earlier in time. All three of these parameters are in balance in these hand-tuned parameter settings.

It is critical to note that, for all of these cases where the hand-tuned parameter settings are in balance, the balance is required for the model to achieve gap and coincidence detection and/or to determine the position of the VOT boundary between categories. This was all the model was designed to do. No parameters were hand-tuned to achieve the other response properties (e.g., asymmetric within-category encoding, onset latency dynamics).

## Analysis of auditory evoked local field potentials

### Identification of key LFP peaks

We identified 3 peaks of the grand mean auditory evoked local field potential (AEP), which were consistent with AEP peaks previously described in the literature (*Howard et al., 2000*; *Nourski et al., 2015*): $P_\alpha$ (positive deflection approximately 75–100 ms after stimulus onset), $N_\alpha$ (negative deflection approximately 100–150 ms after stimulus onset), and $P_\beta$ (positive deflection approximately 150–250 ms after stimulus onset) (see *Figure 1—figure supplements 3* and *4*).

### Bootstrapping approach

For each VOT-sensitive electrode (speech-responsive electrodes whose peak high-gamma amplitude was correlated with VOT), a bootstrapping with resampling procedure was used to estimate the latencies and amplitudes of each peak of the AEP elicited by trials from each VOT condition. During each sampling step in this procedure (1000 bootstrap samples), we computed the average time series of the AEP for each VOT (*Figure 1—figure supplement 4*, panels I-L), the ECoG samples of the time series during each of three time-ranges with the maximum (for positive peaks) or minimum (for the negative peak) mean voltage values for each VOT, and six correlation coefficients (Pearson's *r* between VOT and amplitude/latency for each peak; see *Figure 1—figure supplement 4*, panels M-T).

### Details of peak-finding

$P_\alpha$ was defined as the maximum mean voltage from 0 to 150 ms after stimulus onset, $N_\alpha$ was defined as the minimum mean voltage from 75 to 200 ms after stimulus onset, and $P_\beta$ was defined as the maximum mean voltage from 150 to 250 ms after stimulus onset. To aid peak detection and enforce sequential ordering of the peaks, time ranges for the latter two peaks ($N_\alpha$, $P_\beta$) were further constrained on a per-sample basis by setting the minimum bound of the search time range to be the time of the previous peak (i.e., the earliest possible times for $N_\alpha$ and $P_\beta$ were $P_\alpha$ and $N_\alpha$, respectively). For a given sample, if a peak occurred at either the earliest possible or latest possible time, it was assumed that the peak was either not prominent or did not occur during the defined time range for this electrode/VOT, so that sample was ignored in the analysis for that peak and any subsequent peaks. Because correlation coefficients for each peak were computed over just 6 VOTs in each sample, exclusion of a peak latency/amplitude value for one VOT condition resulted in exclusion of the all conditions for that peak for that sample. Finally, if more than 50% of the bootstrap samples were excluded for a given peak in a given electrode, no samples for that electrode/peak pair were not included in the analysis (see, e.g., $P_\beta$ for e4 in *Figure 1—figure supplement 4*, panels H/P/T).

### Analysis of bootstrapped correlation estimates

For each remaining VOT-sensitive electrode/peak pair, we determined whether or not the latency and/or amplitude of the peak was significantly associated with VOT by evaluating whether the 95% confidence interval (95% CI) across all included bootstrapped estimates of the correlation coefficient excluded 0 (taking the highest density interval of the bootstrapped statistics) (*Figure 1—figure supplement 3*, panel B). These exploratory analyses did not undergo multiple comparison correction.

### Detailed results of analysis of AEPs

The exploratory analyses of correlations between VOT and the latency and/or amplitude of three peaks of the AEP in all VOT-sensitive electrodes revealed four overall conclusions:

1. Comparison of the AEPs evoked by different VOTs shows that there exist associations between stimulus VOT and the amplitude/temporal information in local field potential (LFP). Among electrodes that robustly encode voicing in their peak high-gamma amplitude (i.e.,

VOT-sensitive electrodes), these associations between VOT and LFP features are complex and highly variable (*Figure 1—figure supplement 3*; *Figure 1—figure supplement 4*).

2. Replicating prior results regarding VOT encoding by AEPs (e.g., *Steinschneider et al., 2011*), we find that some electrodes (e.g., e1 in *Figure 1—figure supplement 4*, panels E/I) exhibit temporal encoding of VOT in the latency of various peaks of the AEP. In some electrodes, the nature of this temporal code is straightforward (e.g., in e1, the latency of $N_\alpha$ is delayed by ~10 ms for every additional 10 ms of VOT duration; *Figure 1—figure supplement 4*, panel M), but – more often – the relationship between VOT and peak latency is less direct (*Figure 1—figure supplement 4*, panels N-P).

3. Among electrodes that encode VOT in their peak high-gamma amplitude, there exist many more electrodes that *do not* encode VOT in these temporal features of the AEP (*Figure 1—figure supplement 3*), supporting a prominent role for the peak high-gamma amplitude in the neural representation of voicing and of VOT.

4. Besides the timing of the various AEP peaks, there also exist many electrodes that encode VOT in the amplitude of those peaks (*Figure 1—figure supplement 3*). The encoding patterns are often visually similar to the encoding patterns observed in high-gamma (i.e., graded within the electrode's preferred voicing category; see *Figure 1—figure supplement 4*, panels Q-S). However, there are also many electrodes that do encode VOT in their peak high-gamma amplitude but *not* in these amplitude features of the LFP (*Figure 1—figure supplement 3*, panel B; compare, e.g., *Figure 1—figure supplement 4*, panels D vs. H).

## Supplementary analyses of spatial patterns of VOT effects

Of the 49 VOT-sensitive electrodes, 76% were located posterior to the lateral extent of the transverse temporal sulcus (defined as $y \geq 6$ in MNI coordinate space based on projection of the sulcus onto the lateral STG in the left hemisphere). This is the same region that is densely populated with neural populations that are tuned for other phonetic features (e.g., manner of articulation; *Mesgarani et al., 2014*; *Hamilton et al., 2018*). Mann-Whitney rank-sum tests showed that there was no significant difference in the localization of voiceless-selective (V-) versus voiced-selective (V+) electrodes along either the anterior-posterior axis (*y*-dimension in MNI coordinate space; $U = 342$, $z = -1.23$, p=0.22) or the dorsal-ventral axis (*z*-dimension in MNI coordinate space; $U = 414$, $z = 0.29$, p=0.77).

Although no regional patterns were visually apparent, we tested for hemispheric differences in relative prevalence of VOT-sensitive sites or in voicing category selectivity. Of the seven participants (all of whom had unilateral coverage), four had right hemisphere coverage (57%), and these four patients contributed 28 of the 49 VOT-sensitive electrodes identified in this study (57%) (see *Figure 2A* and *Figure 1—figure supplement 2*; *Supplementary file 1*). Pearson's $\chi^2$ tests confirmed there was no difference in the rate of VOT-sensitive sites ($\chi^2(1)=0.15$, p=0.70) or in the proportion of VOT-sensitive sites that were selective for each category ($\chi^2(1)=1.74$, p=0.19) as a function of hemisphere. Thus, consistent with past ECoG work examining spatial patterns of STG encoding for other phonetic features (e.g., *Hamilton et al., 2018*) we found no evidence that the observed spatial/amplitude code reflected any topographical organization nor any lateralized asymmetries in the encoding of VOT, although data limitations prevent us from ruling out this possibility entirely.

## Acknowledgements

We are grateful to John Houde, who provided the stimuli used in this work, and to all members of the Chang Lab for helpful comments throughout this work. This work was supported by European Commission grant FP7-623072 (MJS); and NIH grants R01-DC012379 (EFC) and F32-DC015966 (NPF). EFC is a New York Stem Cell Foundation-Robertson Investigator. This research was also supported by The William K Bowes Foundation, the Howard Hughes Medical Institute, The New York Stem Cell Foundation and The Shurl and Kay Curci Foundation.

## Additional information

### Funding

| Funder | Grant reference number | Author |
|---|---|---|
| National Institutes of Health | R01-DC012379 | Edward F Chang |
| National Institutes of Health | F32-DC015966 | Neal P Fox |
| European Commission | FP7-623072 | Matthias J Sjerps |
| New York Stem Cell Foundation | | Edward F Chang |
| William K. Bowes, Jr. Foundation | | Edward F Chang |
| Howard Hughes Medical Institute | | Edward F Chang |
| Shurl and Kay Curci Foundation | | Edward F Chang |

The funders had no role in study design, data collection and interpretation, or the decision to submit the work for publication.

### Author contributions

Neal P Fox, Conceptualization, Data curation, Software, Formal analysis, Funding acquisition, Validation, Investigation, Visualization, Methodology; Matthew Leonard, Matthias J Sjerps, Formal analysis, Supervision, Funding acquisition, Validation, Investigation, Visualization, Methodology, Project administration; Edward F Chang, Conceptualization, Supervision, Funding acquisition, Investigation, Project administration

### Author ORCIDs

Neal P Fox (iD) https://orcid.org/0000-0003-0298-3664
Matthew Leonard (iD) https://orcid.org/0000-0002-8530-880X
Edward F Chang (iD) https://orcid.org/0000-0003-2480-4700

### Ethics

Human subjects: All participants gave their written informed consent before surgery and affirmed it at the start of each recording session. The study protocol was approved by the University of California, San Francisco Committee on Human Research. (Protocol number 10-03842: Task-evoked changes in the electrocorticogram in epilepsy patients undergoing invasive electrocorticography and cortical mapping for the surgical treatment of intractable seizures).

### Decision letter and Author response

Decision letter https://doi.org/10.7554/eLife.53051.sa1
Author response https://doi.org/10.7554/eLife.53051.sa2

## Additional files

### Supplementary files

• Supplementary file 1. Table of experimental summary statistics for each participant. Each participant had ECoG grid coverage of one hemisphere (Hem), either left (LH) or right (RH). Participants completed as many trials as they felt comfortable with. Number of trials per participant for ECoG analyses indicate trials remaining after artifact rejection. Some participants chose to listen passively to some or all blocks, so three participants have no trials for behavioral analyses. See Materials and methods for description of inclusion criteria for individual trials in ECoG and behavioral analyses. A subset of speech-responsive (SR) electrodes on the lateral surface of the temporal lobe

had a peak amplitude that was sensitive to VOT, selectively responding to either voiceless (V-) or voiced (V+) stimuli. See Materials and methods for details on electrode selection.

- Supplementary file 2. Table of activation parameters for each model node. $m$ = minimum activation level. $M$ = maximum activation level. $\rho$ = resting activation level. $\lambda$ = decay rate. $\theta$ = propagation threshold.

- Supplementary file 3. Table illustrating timing of 6 simulated model inputs. The table is sparse, meaning that inputs to both Burst and Voicing detector units are 0 whenever a cell is blank. Inputs are clamped onto either Burst or Voicing detector units (always with strength = 1) for a given simulated VOT stimulus during the cycles that are labeled with a B or a V.

- Transparent reporting form

### Data availability

Data and code are available under a Creative Commons License at the project page on Open Science Framework (https://osf.io/9y7uh/).

The following dataset was generated:

| Author(s) | Year | Dataset title | Dataset URL | Database and Identifier |
|---|---|---|---|---|
| Fox NP, Leonard MK, Sjerps MJ, Chang EF | 2020 | Transformation of a temporal speech cue to a spatial neural code in human auditory cortex | https://osf.io/9y7uh/ | Open Science Framework, 9y7uh |

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
