## [Decision Letter]

**Acceptance summary:**

A major challenge for the auditory system is to interpret fine acoustic cues present in the speech signal in order to identify words. This work uses intercranial recordings from human listeners to better understand voice onset time, a key distinction in speech sounds, showing that individual electrodes in temporal lobe are sensitive to voice onset time differences. Complementing experimental work is an example model illustrating how voice onset time might be coded in a neural network.

**Decision letter after peer review:**

Thank you for submitting your article "Transformation of a temporal speech cue to a spatial neural code in human auditory cortex" for consideration by *eLife*. Your article has been reviewed by three peer reviewers, one of whom is a member of our Board of Reviewing Editors, and the evaluation has been overseen by Barbara Shinn-Cunningham as the Senior Editor. The following individual involved in review of your submission has agreed to reveal their identity: Michael Wehr (Reviewer #3).

The reviewers have discussed the reviews with one another and the Reviewing Editor has drafted this decision to help you prepare a revised submission.

Summary

The authors report ECoG data from human listeners while listening to spoken syllables that varied in voice onset time (VOT). They found that VOT (a temporal cue) is encoded in the peak amplitude of high gamma activity in individual electrodes. A simple neural network is presented that qualitatively captures main features of the human data. The findings complement prior ECoG studies on the coding of basic speech properties in the temporal lobes.

Essential revisions

1) The rejection of a temporal code (or temporal contribution) for VOT representation was not entirely convincing. One concern is that the peak window may be too short to capture the temporal dynamics of the signal, so the fact that the classifier fails for temporal information could be artifactual. Examples of the types of dynamics that might temporally encode voicing would be onset latency, peak latency, or waveform shape. The 100 ms peak windows (150-250 ms) miss the onset, probably degrade peak latency information, and likely do not capture the waveform shape (e.g. single or double-peaked, fast or slow rise or decay times, etc). In other words, the HG amplitude appears to be mostly flat during this 100 ms window and thus cannot contain the timing information you wish to test for. Thus the classifier analysis seems almost designed to fail to decode timing information. A different (and possibly more straightforward) way to look at temporal information might be the following. Since you have already extracted the peak time and amplitude, and you want to know whether timing or amplitude convey information, why not just run a classifier on peak times, peak amplitudes, and both? This way instead of removing amplitude information by normalizing, or removing timing information by jittering, you can just directly ask whether the amplitude or timing values can be used to decode voicing. This could serve as a useful corroboration of the multivariate decoding results, or might instead reveal information in peak timing.

2) Although the inclusion of a model was a nice touch, the theoretical contribution of doing so was somewhat unclear. Are there other theoretical frameworks for understanding VOT representations that can be contrasted with the current one? Damper, 1994, is one that was identified by a reviewer (there may be others). Overall we had a difficult time discerning the theoretical advance gained from the model, and a clearer link to existing understandings (does it resolve a controversy?) or clearer way in which it might motivate further experimental approaches would be useful.

3) The focus in the current analysis is on high gamma oscillations. However, other work has suggested a role for low frequency oscillations in phoneme perception (Peelle and Davis 2012; Kösem et al., 2018). So, (a) what's the justification for focusing exclusively on high gamma, and (b) what is a framework for reconciling your high gamma responses with a potential role for lower frequencies?

4) The discussion of local or ensemble temporal coding and spatial coding would benefit from consideration of hierarchical organization and the construction of feature selectivity. If the observed spatial code is the result of some temporal-to-rate transformation, where might this occur and how does that relate to the types of feature selectivity seen in human and primate auditory cortex? As an analogy, your findings are reminiscent of call-echo sensitive cells in the bat. There, many cells in IC respond both to call and to echo (“double-peaked”), whereas other cells in IC respond only to the combination of call and an echo at a particular delay (“single-peaked”). The latter are not topographically organized in IC, but in the FM region of auditory cortex such cells form a topographic map of delay. Do you imagine that a similar hierarchical transformation is occurring in the human auditory system for the encoding of VOT? Where do your recordings and those of e.g. Steinschneider or Eggermont fit into this picture?

5) Please make the stimuli available as supplemental material.

[Editors' note: further revisions were suggested prior to acceptance, as described below.]

Thank you for resubmitting your work entitled "Transformation of a temporal speech cue to a spatial neural code in human auditory cortex" for further consideration by *eLife*. We apologize for the delay in returning a decision to you – due to COVID19-related disruptions to the workflows of many of our editors and reviewers, the process is taking longer than we would like. Thank you for your patience with us.

Your revised article has been evaluated by Barbara Shinn-Cunningham (Senior Editor) and a Reviewing Editor. We agree that the manuscript has been improved but there are some remaining issues that need to be addressed before acceptance. Specifically, the issue of the distinction between the hypotheses laid out in subsection “Peak neural response amplitude robustly encodes voicing category”.

The issues are nicely laid out by reviewer 3's comments, which we include unedited (so will not repeat here). It seems clear that the manuscript makes a valuable contribution and we have no remaining major issues with the analyses. However, it will be important to address the apparent contradictions in the results and conclusions in discriminating between the hypothesses laid out in subsection “Peak neural response amplitude robustly encodes voicing category”.

Of course, there is a chance we all missed something obvious – please let us know, and perhaps some clarification in the text would be helpful in this case.

Reviewer #3:

The authors have done a good job addressing the comments and the revised manuscript is responsive to most of the points raised. The additional interpretation of the model results, robustness, and context are welcome additions. The discussion of coding and hierarchical processing are also good. Yet there is a remaining issue that sticks out and that needs to be resolved. This is the question of whether the temporal patterns of neural responses encode VOT information. To be clear, I don't have a dog in this fight – I'm neutral about spatial or temporal codes. I'm just pointing out that the manuscript is internally conflicted on this point, and the revisions haven't resolved the issue.

As the authors spell out in the rebuttal, "It is certainly the case that both sub-categorical and category-level information is carried by the onset latency of voiced-selective (V+) neural populations (Figure 3). However, this temporal information does not contribute to classification of voicing category (Figure 1F) because this information is not available during the peak window." Reading the reporting of the amplitude/timing decoding shown in Figure 1F, the take-home message from that is that peak amplitude, but not timing, contains VOT information. This message is wrong, because as shown in Figure 3 the onset latency encodes VOT information. So care must be taken to avoid leading readers towards that message.

Close reading of the Results section reporting Figure 1F reveals that the statements are accurate because they contain a clause such as "in the peak response window," for example: "In contrast, when amplitude information was corrupted and only temporal patterns in the peak response window were reliable (-Amplitude/+Timing), classifier performance was not different from chance." Even though this statement is accurate, I'd argue that it's misleading, especially because the set-up is to distinguish between 3 hypotheses: "Specifically, we evaluated three alternatives for how temporally-cued voicing category is encoded by high-gamma responses in cortex: (1) the spatial pattern of peak response amplitude across electrodes, (2) the temporal patterns of evoked responses across electrodes, or (3) both amplitude and timing of neural activity patterns." At the end of this section, after looking at Figure 1F, the reader is left with hypothesis (1) as the take-home. But your data rule out (1) and instead demonstrate hypothesis (3), but not until Figure 3. I get the motivation that you want to show encoding by peak amplitude in order to compare with previous findings from your group. That's fine. But there's no need to rule out a temporal code to do this. If the take-home message from Figure 1 is that VOT information is encoded in peak amplitude, a spatial code, just say that, and drop the temporal jitter analysis, because it's misleading and unnecessary. Or else expand the window to include onsets, which based on Figure 3 should support VOT classification.

---

## [Author Response]

Essential revisions1) The rejection of a temporal code (or temporal contribution) for VOT representation was not entirely convincing. One concern is that the peak window may be too short to capture the temporal dynamics of the signal, so the fact that the classifier fails for temporal information could be artifactual. Examples of the types of dynamics that might temporally encode voicing would be onset latency, peak latency, or waveform shape. The 100 ms peak windows (150-250 ms) miss the onset, probably degrade peak latency information, and likely do not capture the waveform shape (e.g. single or double-peaked, fast or slow rise or decay times, etc). In other words, the HG amplitude appears to be mostly flat during this 100 ms window and thus cannot contain the timing information you wish to test for. Thus the classifier analysis seems almost designed to fail to decode timing information. A different (and possibly more straightforward) way to look at temporal information might be the following. Since you have already extracted the peak time and amplitude, and you want to know whether timing or amplitude convey information, why not just run a classifier on peak times, peak amplitudes, and both? This way instead of removing amplitude information by normalizing, or removing timing information by jittering, you can just directly ask whether the amplitude or timing values can be used to decode voicing. This could serve as a useful corroboration of the multivariate decoding results, or might instead reveal information in peak timing.

We understand the concerns embodied by this reviewer comment and appreciate the suggestions offered here. We have addressed them in four ways.

First, we have revised and clarified our claims to state that we are *not* ruling out a temporal code entirely, but instead are focusing on our key, novel result: the highly robust encoding of voicing in the peak neural response amplitude. For the reasons the reviewer mentioned, it is difficult, if not impossible, to completely rule out any possible role for a temporal code, since there are many possible ways in which such a code could manifest (including several that are discussed here).

What we find most important and striking are:

a) that the encoding of voicing category on single trials appears to depend strongly on the response amplitude during the peak window, but does not seem to depend greatly on the fine temporal dynamics during that window, and

b) that the signals that encode VOT in their amplitude (namely, the peaks of high gamma evoked responses) are the same as the signals that have previously been shown to encode other (primarily spectrally-cued) phonetic features (Mesgarani, Cheung, Johnson, and Chang, 2014).

We have made numerous modifications throughout the manuscript to clarify what our results do and do not show (Results and Discussion). Overall, we have clarified the conclusions we draw from the decoding analyses depicted in Figure 1F in order to focus on the above claims, which we believe constitute novel and important results.

Second, we have clarified the rationale for our original analysis approach. To that end, we address several specific points in the reviewer comment:

Peak window: The analysis of amplitude and temporal information was designed to be confined to the peak response window (150-250 ms after stimulus onset), a time window of interest to us based on prior work examining the encoding of spectrally cued phonetic features within the peak high-gamma responses of spatially discrete neural populations in human temporal lobe (Mesgarani et al., 2014). We do not believe that, *a priori*, this makes the temporal model likely to fail.

Waveform shape during peak: Although the amplitude may appear to be mostly flat during this window in the trial-averaged traces shown in Figure 1E, they are by no means flat on a single-trial basis (see Author response image 1). Indeed, our peak window also overlaps almost entirely with a window during which past work (examining intracranial auditory evoked local field potentials) found evidence of waveform shape differences between voiced and voiceless stimuli (single- vs. double-peaked responses), prompting claims of temporal coding of VOT (see, e.g., Figure 10 of Steinschneider et al., 2011). In other words, there was no *a priori* reason to believe that the peak window we selected would not also contain temporal information in the form of waveform shape differences. If such reliable differences had existed in our high-gamma data, our method of corrupting temporal information (jittering) would have detected a contribution of temporal information to decoding, but it did not. Indeed, the fact that the trial-averaged waveforms appear to be relatively flat during this window (even though single-trial waveforms are not) is visual evidence that waveform shape is not a reliable cue to voicing category here.

Peak latency: To the extent that the peak of a trial’s evoked high-gamma response occurs during or close to the peak window, the contribution of peak latency information to decoding accuracy would also be captured by our approach, as the temporal jittering procedure would disrupt the reliability of this information. We address this issue directly in a new analysis described below (see Author response images 1 and 2).

Onset latency: If the only difference between the high-gamma responses elicited by different VOTs was when the response started (i.e., its onset latency), with all other aspects of the waveform’s shape remaining constant across conditions (i.e., a “phase shift”), there would also be reliable VOT-dependent differences in the responses’ temporal dynamics during the peak response window.

Other possible sources of temporal codes:

Onset latency: As is evident from our results in Figure 3, there are reliable VOT dependent temporal differences among response onset latencies in voiced selective electrodes that are apparently not reflected during the peak window (since decoding is not significantly affected by temporal jittering).

Outside of high-gamma range: Despite our focus on the high-gamma range, we recognize the importance of potential reliable temporal coding features carried by other components of the neural response, such as lower-frequency components. We address this possibility directly in new supplementary analyses (discussed in response to Essential Revision #3 in this letter; see Figure 1—figure supplements 3 and 4 in revised manuscript).

We have revised the manuscript to clarify the types of temporal information that would be corrupted by the temporal jittering approach (Results and Materials and methods). We also now emphasize that the results of our decoding analysis serve primarily to highlight the contribution of peak high-gamma amplitude to VOT representation (a novel result), but this analysis cannot elucidate whether other temporal properties of the neural response could also carry information about VOT (e.g., outside of the peak window or outside of the high-gamma range) (Results).

Third, we conducted the analysis suggested by the reviewer(s), obtaining results that ultimately support the same conclusion as our original decoding/classifier analyses. In Author response image 1, we illustrate what the data look like for two representative electrodes (one voiceless-selective [e1] and one voiced-selective [e2]) by plotting the high-gamma traces elicited by each VOT stimulus on six individual trials (Author response image 1, left). For each trial, the peak high-gamma activity was identified. Next, we plot the peak latency and amplitude for every trial for each of the example electrodes in order to illustrate the clearer separation of VOT categories using the amplitude information (Author response image 1, right).

**Author response image 1. respfig1:** Single electrodes demonstrate better separation of voicing category based on peak amplitude vs. peak latency. left: High-gamma traces for six single trials (one per VOT condition, as indicated by line color; 0ms VOT = red; 50ms VOT = blue; example trials shown for visual simplicity) in each of two example VOT-sensitive electrodes (e1: voiceless-selective; e2: voiced-selective; same electrodes as shown in Figure 1 of the main text). Black dots indicate the peak high-gamma amplitude and latency for each trial. There is clear variation among single trials in the peak’s timing and amplitude. middle: The latency of the peak, tp, (in seconds) for each trial (n = 234 total trials; color of circles corresponds to trial’s voicing category: /b/ = red; /p/ = blue) projected into a 2-dimensional space, with the vertical and horizontal dimensions representing the two example electrodes (e1 vs. e2). Trials were selected such that peaks occurred between 0 and 0.5 seconds after stimulus onset. This panel illustrates the lack of a reliable difference between voicing categories based on the peak latency. right: The amplitude of the peak, HGz(tp), for each trial projected into the same 2-dimensional space illustrates the highly reliable difference between voicing categories based on peak amplitude.

Per the reviewer’s suggestion, we quantified these results by building two separate classifiers (linear discriminant analysis with leave-one-out cross-validation) that used either peak latency or peak amplitude information separately. Consistent with our original result, we found that peak latency information alone did not lead to above-chance accuracy, while peak amplitude information performed significantly better than chance, and also significantly higher than the peak latency classifier (Author response image 2).

**Author response image 2. respfig2:** Peak amplitude outperforms peak latency in classifying single-trial voicing category. For each participant, two classifier analyses were conducted to predict each trial’s voicing category using leave-one-out cross-validation. All speech-responsive electrodes for a given patient were included in both classifiers, but classifiers included only either temporal [Temp; peak latency = tp] or amplitude [Amp; peak amplitude = HGz(tp)] features. Across participants, only amplitude features performed better than chance (chance = 50%), and amplitude features performed significantly better than temporal features (ps < 0.01; Wilcoxon signed-rank tests). Error bars represent standard error across participants.

As discussed above, we believe that our original decoding analysis constitutes a more general test of the hypothesis that peak amplitude information is a robust predictor of voicing category because peak latency information is just one type of temporal information that is included in our original decoding analysis, along with waveform shape, which could not be captured in this alternative analysis of single-trial peaks. Since both analyses ultimately point to the same conclusion, but the original analysis is more general, and since we have revised our claims to focus less on rejecting temporal codes than on illustrating the robustness of the amplitude information, we have opted to leave the original decoding analysis in the manuscript.

Please note, however, that if the Editors and reviewers feel it would help strengthen the manuscript, we are happy to include Author response image 1 and Author response image 2 as figure supplements in the final manuscript.

Fourth, and in line with our renewed focus and recognition that it is not possible to completely reject every potential role for temporal information in VOT perception or representation, we present another new analysis that examines temporal coding features not contained within the high-gamma response. Because this new analysis is also responsive to other reviewer comments regarding contributions of lower-frequency information to VOT encoding, we discuss it in detail in response to Essential Revision #3 in this letter (see Figure 1—figure supplements 3 and 4 in revised manuscript). Regarding Essential Revision #1, though, the most relevant update to the manuscript is an acknowledgement that this result demonstrates that temporal and amplitude representations are not mutually exclusive (Discussion).

2) Although the inclusion of a model was a nice touch, the theoretical contribution of doing so was somewhat unclear. Are there other theoretical frameworks for understanding VOT representations that can be contrasted with the current one? Damper, 1994, is one that was identified by a reviewer (there may be others). Overall we had a difficult time discerning the theoretical advance gained from the model, and a clearer link to existing understandings (does it resolve a controversy?) or clearer way in which it might motivate further experimental approaches would be useful.

We appreciate this question, and are eager to clarify the role we believe the model plays in this study. We had two primary goals in including a computational model: (1) using simple, theoretically-motivated, and well-established computational mechanisms, we wanted to replicate as many of the key aspects of our data as possible *in silico*; and (2) we wanted to provide a mathematical description of our key result, namely that a temporal speech cue is encoded by a spatial (amplitude) code across different neural populations.

As with most computational modeling approaches, there were a large number of possible architectures and algorithms we could have chosen, including the one mentioned by the reviewers (Damper, 1994). Here, we were guided primarily by *Occam’s Razor*, seeking to implement an extremely simple model that could be linked directly to the scale of the neural data we have available (namely, population-level ECoG electrodes).

Motivated by previous literature (Buonomano and Merzenich, 1995; Carr, 1993; Gao and Wehr, 2015; Rauschecker, 2014), we sought to implement two types of computation that are well-established and reasonable hypotheses for the observation that some electrodes are voiceless-selective and others are voiced-selective. Specifically, the model demonstrates that the key findings regarding the encoding of a temporal speech cue in the amplitude of the peak neural response at spatially discrete neural populations emerge naturally from the time-dependent mechanisms of a simple neural network model with coincidence- and gap-detector circuits. If we had not actually implemented this model, we would have been forced to speculate about how these (or other) computations could underlie the observed data in the Discussion. Instead, we believe that the model allows us to go beyond pure speculation, both in providing an implemented mathematical explanation, and in providing a framework for generating explicit, testable hypotheses for future follow-up work. Thus, although the model we present was not specifically designed to resolve a controversy or distinguish between two particular competing hypotheses about VOT perception, we believe that these theoretical contributions are significant and stand on their own merits.

We also think it is important that the model we created captures aspects of the neural activity that were not explicitly designed into the model itself. This simple architecture that can achieve gap and coincidence detection also predicts the observed partially-graded within-category encoding of VOT. Additionally, the early temporal dynamics at these spatially localized cortical sites are also predicted by the model. This final point is particularly important because the early temporal dynamics were never considered when selecting electrodes for inclusion in the study, but were perfectly in line with the model’s predictions.

Together, we believe that these motivations and results warrant including the neural network model. It helps us achieve an important theoretical contribution by providing an explicit, testable account that connects multiple seemingly disparate observations about the neurophysiological data. In fact, all of the complex encoding properties we observed arise directly from a simple model designed to perform gap/coincidence detection by implementing theoretically-motivated, well-established neuronal circuits.

We have added text to the Discussion to clarify the motivation and theoretical contributions of the model. In addition, we agree with the reviewers that it is important to provide additional context for the specific model we chose; therefore, we have contextualized our model among other possible approaches (Damper, 1994; McClelland and Elman, 1986) within the Discussion.

3) The focus in the current analysis is on high gamma oscillations. However, other work has suggested a role for low frequency oscillations in phoneme perception (Peelle and Davis 2012; Kösem et al., 2018). So, (a) what's the justification for focusing exclusively on high gamma, and (b) what is a framework for reconciling your high gamma responses with a potential role for lower frequencies?

We thank the reviewer for bringing up these important points. We will respond to each of them separately:

a) Our primary goal was to examine neural responses within the temporal lobe to stimuli varying in their voice-onset time (VOT) using the same neural features previously used to illustrate a spatial (amplitude) code for other phonetic features, including manner and place of articulation of consonants (Mesgarani et al., 2014). In our view, this is critical because it seeks to unify two lines of prior research examining neurophysiological representations of phonetic features cued primarily by spectral acoustic information (e.g., manner/place of articulation) or primarily by temporal acoustic information (e.g., voicing). Up to now, there have been few attempts to address the central theoretical question of whether a common neural code for both exists. Some prior work (especially, but by no means exclusively, work examining auditory evoked local field potentials in primary auditory cortex) has posited that the neural code for VOT differs fundamentally from the code for spectrally-cued phonetic features, with only the latter relying on a spatial code (see, e.g., Steinschneider, Nourski, and Fishman, 2013; Steinschneider, Volkov, Noh, Garell, and Howard, 1999). Meanwhile, the above-referenced recent demonstration of a robust spatial code for spectrally-cued phonetic features focused specifically on the peak high-gamma response amplitude of neural populations in human superior temporal gyrus (Mesgarani et al., 2014).

Here, we explicitly tested the hypothesis that the same encoding scheme is used to represent phonetic features defined primarily by temporal information. To that end, we focused on stimulus-evoked activity in the high-gamma range of the neural response. Additionally, particularly for direct intracranial recordings, while there is a relatively clear link between high-gamma activity and neuronal firing, the underlying sources and single-/multi-unit activity that give rise to lower frequencies and oscillations are less well-understood. Therefore, while we do not deny the important roles of lower frequency activity, we believe we can make the clearest and most interpretable neurophysiological claims based on intracranially-recorded high-gamma.

b) Our work does not discount a potential role for low-frequency oscillations in speech perception or in the perception of phonemes. Indeed, our results are not inconsistent with the large body of work focused on phase-amplitude coupling between low frequency oscillations and gamma power (e.g., Fries, 2009; Giraud and Poeppel, 2012; though note that these frameworks typically refer to power in a lower gamma band than is used here). Specifically, it is possible that our perception of voicing based on VOT information (or of other temporally-cued phonetic features) may also depend on or interact with lowfrequency oscillations (Kösem et al., 2018; Peelle and Davis, 2012). These signals may, in fact, be coupled in their phase-amplitude relationship, and, according to the theoretical frameworks in that body of work, it may be the case that low frequency phase information modulates firing rates observed in higher frequency broadband activity.

In our opinion, a detailed examination of the relationship between low-frequency amplitude and/or phase and high-gamma power and their contributions to VOT encoding in speech is beyond the scope of the current manuscript, since our primary goal was to examine the encoding of VOT using a signal which has been shown to encode other phonetic features. However, we agree with the reviewer(s) that it is important to address these same questions using signals that have been used in directly related work (e.g., work by Steinschneider and Nourski).

To that end, we have now conducted an additional analysis of the neural responses to our stimuli using the raw voltage local field potential (LFP), which is dominated by lower frequency components. For every VOT-sensitive electrode identified in our study, we used a bootstrapping approach to analyze the correlation between VOT and the peak latency and amplitude of 3 peaks in the auditory evoked potential (AEP): *P*_a_, *N*_a_, *P*_b_ (Howard et al., 2000; Nourski et al., 2015). Detailed descriptions of these analyses and their results now appear in new subsections of Materials and methods.

We also summarize the results of these additional analyses in the main text (Results). Two new figure supplements (Figure 1—figure supplements 3 and 4) illustrate the following four conclusions:

1) Comparison of the AEPs evoked by different VOTs shows that there exist associations between stimulus VOT and the amplitude/temporal information in the LFP. Among electrodes that robustly encode voicing in their peak high-gamma amplitude (i.e., VOT-sensitive electrodes), these associations between VOT and LFP features are complex and highly variable (Figure 1—figure supplements 3 and 4).

2) Replicating prior results regarding VOT encoding by AEPs (e.g., Steinschneider et al., 2011), we find that some electrodes (e.g., e1 in Figure 1—figure supplement 4, panels E/I) exhibit temporal encoding of VOT in the latency of various peaks of the AEP. In some electrodes, the nature of this temporal code is straightforward (e.g., in e1, the latency of *N*_a_ is delayed by ~10ms for every additional 10ms of VOT duration; Figure 1—figure supplement 4, panel M), but – more often – the relationship between VOT and peak latency is less direct (Figure 1—figure supplement 4, panels N-P).

3) Among electrodes that encode VOT in their peak high-gamma amplitude, there exist many more electrodes that *do not* encode VOT in these temporal features of the AEP (Figure 1—figure supplement 3, panel B), supporting a prominent role for the peak high-gamma amplitude in the neural representation of voicing and of VOT.

4) Besides the timing of the various AEP peaks, there also exist many electrodes that encode VOT in the amplitude of those peaks (Figure 1—figure supplement 3, panel B). The encoding patterns are often visually similar to the encoding patterns observed in high-gamma (i.e., graded within the electrode’s preferred voicing category; see Figure 1—figure supplement 4, panels Q-S).

We feel that connecting our data to the previous literature with these additional analyses has substantially enhanced the contribution of our work. Besides these additional analyses, and in response to this and other reviewer comments, we have also updated the manuscript to clarify and emphasize the goal of our study (Results), and to acknowledge potential roles for low-frequency components of the neural response in the perceptual experience of speech and in its neurophysiological representation (Discussion), as discussed above.

Ultimately, we hope that the revised manuscript communicates that there is interesting and important information carried within lower frequencies (and, in some cases, by their temporal dynamics), while also emphasizing what we view as the significant theoretical contribution constituted by our robust, novel high-gamma data, which connect directly to previous findings regarding speech sound encoding (Discussion). In contrast to prior work theorizing parallel, but fundamentally different, coding schemes for spectrally- and temporally-cued phonetic features, we demonstrate evidence for a shared representation of both by high-gamma in the human superior temporal lobe.

4) The discussion of local or ensemble temporal coding and spatial coding would benefit from consideration of hierarchical organization and the construction of feature selectivity. If the observed spatial code is the result of some temporal-to-rate transformation, where might this occur and how does that relate to the types of feature selectivity seen in human and primate auditory cortex? As an analogy, your findings are reminiscent of call-echo sensitive cells in the bat. There, many cells in IC respond both to call and to echo (“double-peaked”), whereas other cells in IC respond only to the combination of call and an echo at a particular delay (“single-peaked”). The latter are not topographically organized in IC, but in the FM region of auditory cortex such cells form a topographic map of delay. Do you imagine that a similar hierarchical transformation is occurring in the human auditory system for the encoding of VOT? Where do your recordings and those of e.g. Steinschneider or Eggermont fit into this picture?

We thank the reviewers for raising this important question. We believe that this question highlights an important point: that temporal gap detection is a pervasive mechanism in neural processing of auditory stimuli and that coincidence and gap detection can (and likely does) arise at many levels of the nervous system.

Unfortunately, we do not think we can make strong claims about the hierarchical organization of this type of coding, since the recordings conducted as part of this study do not include data from either subcortical areas (like the inferior colliculus) or primary auditory cortex (Heschl’s gyrus). Therefore, while we agree with the reviewer that the nature of the hierarchical encoding of temporal cues is an important issue and would also link directly to other work in both animal models and humans, most of what we can say would be speculation.

That said, while we do not wish to speculate too much on these topics, we have addressed the important issues raised by this comment in three ways in the revised manuscript (Discussion).

First, our model actually suggests that temporal integration may be occurring locally. The inputs to the gap and coincidence detectors in the model are only spectrally processed (burst and voicing detectors), which arrive at various temporal latencies to the coincidence and gap detector units (which are meant to directly represent neural populations in the STG). As such, the model’s prediction of the within-category patterns (Figures 2B-D) and (especially) the temporal onset latency dynamics (Figure 3) are consistent with local temporal integration rather than gap and coincidence detection that is inherited from earlier levels of processing (e.g., from midbrain processing). However, we recognize that this is not a definitive interpretation, and, more importantly, even a finding that temporal integration is occurring locally in non-primary auditory cortex does not preclude that temporal integration could be simultaneously occurring at other (lower) levels of the ascending auditory pathway, including in IC. We have summarized this response in the Discussion.

Second, it is also worth noting that, contrary to the topographic map of delay described in the FM region of bats, neither the present study nor any others that we are aware of offer evidence of a topographic map of VOT encoding, nor of any other phonetic features, in human superior temporal gyrus. Therefore, the analogy to these animal models may be incomplete, and may require further direct work. This point is now summarized in the Discussion, and additional results regarding the lack of any discernible topographic organization are described in Materials and methods.

Third, although our discussion of hierarchical transformations in auditory representations is limited, we have tried to clarify how our results relate to past work (e.g., work by Steinschneider and Eggermont mentioned in the reviewer comment) by conducting and reporting new analyses of auditory evoked local field potentials, as described in our response to Essential Revision #3.

5) Please make the stimuli available as supplemental material.

We agree that this addition will enhance the contribution of our study, and have included the stimuli among the supplementary materials (see Materials and methods).

[Editors' note: further revisions were suggested prior to acceptance, as described below.]

Reviewer #3:The authors have done a good job addressing the comments and the revised manuscript is responsive to most of the points raised. The additional interpretation of the model results, robustness, and context are welcome additions. The discussion of coding and hierarchical processing are also good. Yet there is a remaining issue that sticks out and that needs to be resolved. This is the question of whether the temporal patterns of neural responses encode VOT information. To be clear, I don't have a dog in this fight – I'm neutral about spatial or temporal codes. I'm just pointing out that the manuscript is internally conflicted on this point, and the revisions haven't resolved the issue.As the authors spell out in the rebuttal, "It is certainly the case that both sub-categorical and category-level information is carried by the onset latency of voiced-selective (V+) neural populations (Figure 3). However, this temporal information does not contribute to classification of voicing category (Figure 1F) because this information is not available during the peak window." Reading the reporting of the amplitude/timing decoding shown in Figure 1F, the take-home message from that is that peak amplitude, but not timing, contains VOT information. This message is wrong, because as shown in Figure 3 the onset latency encodes VOT information. So care must be taken to avoid leading readers towards that message.Close reading of the Results section reporting Figure 1F reveals that the statements are accurate because they contain a clause such as "in the peak response window," for example: "In contrast, when amplitude information was corrupted and only temporal patterns in the peak response window were reliable (-Amplitude/+Timing), classifier performance was not different from chance." Even though this statement is accurate, I'd argue that it's misleading, especially because the set-up is to distinguish between 3 hypotheses: "Specifically, we evaluated three alternatives for how temporally-cued voicing category is encoded by high-gamma responses in cortex: (1) the spatial pattern of peak response amplitude across electrodes, (2) the temporal patterns of evoked responses across electrodes, or (3) both amplitude and timing of neural activity patterns." At the end of this section, after looking at Figure 1F, the reader is left with hypothesis (1) as the take-home. But your data rule out (1) and instead demonstrate hypothesis (3), but not until Figure 3. I get the motivation that you want to show encoding by peak amplitude in order to compare with previous findings from your group. That's fine. But there's no need to rule out a temporal code to do this. If the take-home message from Figure 1 is that VOT information is encoded in peak amplitude, a spatial code, just say that, and drop the temporal jitter analysis, because it's misleading and unnecessary. Or else expand the window to include onsets, which based on Figure 3 should support VOT classification.

We thank reviewer 3 for these comments. The reviewer is correct that all claims referencing Figure 1 (including the panel in question – Figure 1F) are meant to apply only to the peak response window. Indeed, we believe that one of the primary contributions of this work is to show that peak high-gamma amplitude robustly encodes voicing category. Figure 1F shows that spatially distributed amplitude patterns are a robust code during the peak response window (150-250ms after stimulus onset) irrespective of whether or not timing information is corrupted.

As reviewer 3 acknowledges, and as we point out in the manuscript, this peak time window is of special interest because past work has shown that response amplitude of some neural populations throughout STG during this critical window constitute a spatial code for other phonetic properties of speech sounds (e.g., manner/place of articulation). Our primary goal was to test whether a temporally-cued phonetic distinction (voicing/VOT) might be represented within the same neural coding scheme, and our work shows that stop consonant voicing can, in fact, also be accounted for within this same theoretical framework.

We believe that the robustness of a spatial/amplitude code during this peak time window will be of great interest to readers of this paper, and so have opted not to remove these analyses. Instead, we have added clarifying language emphasizing that our results in Figure 1F refer only to the (critically interesting) peak neural response window (see revisions outlined below).

As reviewer 3 notes, subsequent analyses showed that sub-categorical and category level information is carried by the onset latency of voiced-selective (V+) neural populations (Figure 3). To better integrate the results in Figure 1 and Figure 3, we have also added text to point the reader to this secondary result and highlight the fact that it is in no way contradictory with our primary result (the spatial/amplitude code for voicing during the peak window).

In order to address this comment, we have made changes in several places in the manuscript:

1) Figure 1 caption’s title:

a) “Speech sound categories that are distinguished by a temporal cue are spatially encoded in the peak amplitude of neural activity in distinct neural populations.”

2) Motivation of classifier analyses shown in Figure 1F:

a) “As with the previous analyses, and following prior work on speech sound encoding, these analyses (Figure 1F) focused on cortical high-gamma activity during the peak response window (150-250ms after stimulus onset; but see Figure 3 for analyses of an earlier time window).”

3) Addition of language emphasizing that the analyses in Figure 1F apply only to the “peak response window”

4) Caveats pointing the reader to Figure 3 for evidence of temporal encoding patterns:

a) “Note that, while spatial (and not temporal) patterns of high-gamma responses robustly encode voicing during this critical peak window, we later describe additional analyses that address possible temporal encoding patterns in the local field potential (Figure 1—figure supplements 3 and 4) and in an earlier time window (Figure 3).” (Results)

b) “…clearly demonstrating that temporal and amplitude codes for VOT are not mutually exclusive (see also temporal encoding patterns in onset latencies of V+ electrodes; Figure 3)” (Discussion)

5) Clear interpretation of Figure 3 as evidence of temporal encoding pattern when looking outside of the peak response window:

a) “Finally, Figure 3 shows that, unlike during the peak response window (150250ms after stimulus onset; Figure 1F), temporal information does encode VOT during an earlier window around the neural response onset in some neural populations. Indeed, both sub-phonetic and phonetic category-level information are carried by the onset latency of V+ electrodes, with evoked responses arising later at these sites for stimuli with progressively longer VOTs. Critically, the modeling results indicate that both the amplitude encoding patterns during the peak window and the temporal encoding patterns during the earlier onset window are captured by the same canonical neurophysiological mechanisms.”